# Hasty sensorimotor decisions rely on an overlap of broad and selective changes in motor activity

**Gerard Derosiere**[1] *, **David Thura**[2], **Paul Cisek**[3], **Julie Duque**[1]

1 Institute of Neuroscience, Laboratory of Neurophysiology, Université Catholique de Louvain, Brussels, Belgium, 2 Lyon Neuroscience Research Center–Impact Team, Inserm U1028, CNRS UMR5292, Lyon 1 University, Bron, France, 3 Department of Neuroscience, Université de Montréal, Montréal, Canada

* gerard.derosiere@uclouvain.be

**Data Availability Statement:** All data and code will be fully available from the OSF database at: https://osf.io/tbw7h/.

**Funding:** JD was supported by a grant from the Belgian National Funds for Scientific Research

## Abstract

Humans and other animals are able to adjust their speed–accuracy trade-off (SAT) at will depending on the urge to act, favoring either cautious or hasty decision policies in different contexts. An emerging view is that SAT regulation relies on influences exerting broad changes on the motor system, tuning its activity up globally when hastiness is at premium. The present study aimed to test this hypothesis. A total of 50 participants performed a task involving choices between left and right index fingers, in which incorrect choices led either to a high or to a low penalty in 2 contexts, inciting them to emphasize either cautious or hasty policies. We applied transcranial magnetic stimulation (TMS) on multiple motor representations, eliciting motor-evoked potentials (MEPs) in 9 finger and leg muscles. MEP amplitudes allowed us to probe activity changes in the corresponding finger and leg representations, while participants were deliberating about which index to choose. Our data indicate that hastiness entails a broad amplification of motor activity, although this amplification was limited to the chosen side. On top of this effect, we identified a local suppression of motor activity, surrounding the chosen index representation. Hence, a decision policy favoring speed over accuracy appears to rely on overlapping processes producing a broad (but not global) amplification and a surround suppression of motor activity. The latter effect may help to increase the signal-to-noise ratio of the chosen representation, as supported by single-trial correlation analyses indicating a stronger differentiation of activity changes in finger representations in the hasty context.

## Introduction

From insects to rodents to primates, sensorimotor decisions are characterized by an inherent covariation between speed and accuracy [1–3], making the speed–accuracy trade-off (SAT) a universal property of animal behavior [4,5]. Still, humans and other animals are able to adjust their SAT at will depending on the urge to act, favoring either hasty (i.e., high speed and low accuracy) or cautious (i.e., low speed and high accuracy) decision policies in different contexts.

(FNRS: MIS F.4512.14). GD was a postdoctoral fellow supported by the FNRS (FNRS: 1B134.18). The funders had no role in study design, data collection and analysis, decision to publish, or preparation of the manuscript.

**Competing interests:** The authors have declared that no competing interests exist.

**Abbreviations:** ADM, abductor digiti minimi; ANCOVA, analysis of covariance; APB, abductor pollicis brevis; BF, Bayes factor; CI, confidence interval; DT, decision time; EEG, electroencephalogram; EMG, electromyography; FDI, first dorsal interosseous; fMRI, functional magnetic resonance imaging; GG, Greenhouse–Geisser; HSD, honestly significant difference; LG, lateral gastrocnemius; M1, primary motor cortex; MEP, motor-evoked potential; MG, medial gastrocnemius; MSO, maximum stimulator output; rmANOVA, repeated measures analyses of variance; rmCorr, repeated measures correlation; rMT, resting motor threshold; RT, reaction time; SAT, speed–accuracy trade-off; SCA, seed-based correlation analysis; SD, standard deviation; SRT, simple RT; TA, tibialis anterior; TMS, transcranial magnetic stimulation.

Given the importance of SAT regulation in the animal realm, and the deleterious impact of its disruption in major human diseases, such as in impulse control disorders [6–9], extensive research is being devoted to understanding its neural basis [10–12].

Sensorimotor theories of decision-making postulate that decisions between actions arise, at least partly, from a competition between neural populations responsible for action execution in the motor system [13–20]. This theoretical view has prompted the field to investigate the motor system as a potential site for SAT regulation [1–3,21–27]. Consistently, motor activity appears to undergo influences pulling it upward when the urge to act is high, in contexts calling for hasty decisions [1–3,21,26,27]. Furthermore, converging lines of evidence suggest that the source of this modulation may involve subcortical structures, especially the basal ganglia [25,27–30] and the noradrenergic system [1,2,31,32], which are known to exert broad influences on the motor cortex. Because of these 2 sets of findings, an emerging view in the field is that SAT regulation relies on influences exerting broad changes in the motor system [1–3], tuning its activity up in a global manner when hastiness is at premium, irrespective of the neural population ultimately recruited for the action.

Global modulation represents a key candidate mechanism for how animals adjust their behavior in different SAT contexts, especially when considered in the light of computational models of decision-making [10,33]. In "drift-diffusion models," deliberation between actions involves an accumulation of evidence, which drives the buildup of neural signals coding for different actions toward a critical decision threshold in the motor system, and once one of them reaches this threshold, the related action is chosen and executed [34–42]. An alternative model suggests that sensory evidence is computed quickly, and the buildup of neural signals is primarily due to a growing "urgency signal" that pushes the system to reach the decision threshold even if evidence remains low [43–45]. While there is continued debate on whether neural activity buildup is primarily due to evidence accumulation versus urgency, all of these models suggest that control of SAT can be accomplished through a global motor amplification (i.e., in hasty contexts). This unique mechanism would explain how animals speed up their decisions and why they are more prone to make incorrect choices when they do so.

The explanatory power of the global modulation idea has contributed to its dissemination in the field of decision-making [1]. Yet, direct evidence for a context-dependent modulation of activity that is global across the motor system remains scarce in the SAT literature. In fact, if changes in motor activity have been interpreted through the lens of a global mechanism, the studies themselves were not designed to address directly the scope of modulatory changes per se, which would require considering different neural populations across the somatotopic map. Indeed, single-cell studies in monkeys only targeted one particular population (e.g., the arm area during reaching decisions [3,24] or eye areas during oculomotor decisions [26,27]), while electroencephalography studies in humans lacked the spatial resolution for tackling this critical issue [1,2]. A current challenge in the field is thus to characterize the scope of activity changes that may occur in different neural populations of the motor system during SAT regulation. Addressing this critical issue would provide insights into which structures may be at the origin of SAT-based motor changes and how neuromodulation of motor activity may be used to adjust SAT in impulse control disorders that are typically associated with hasty behaviors.

One fruitful approach to tackle this challenge is through the analysis of motor-evoked potentials (MEPs), elicited by the application of single-pulse transcranial magnetic stimulation (TMS) over the primary motor cortex (M1) [46,47]. When applied over M1, TMS depolarizes populations of corticospinal neurons—often referred to as "motor representations"—which project down to specific body parts [48,49] and generates MEPs in targeted muscles. Because corticospinal populations partly overlap in M1, TMS applied over one site can elicit MEPs in several muscles that are close together (e.g., several finger muscles). Importantly, neurons of

the corticospinal pathway are under the influence of various subcortico-cortical circuits [50]. Hence, the amplitude of MEPs provides a population-specific readout of the net impact of these modulatory circuits on motor representations at the time of the stimulation [51].

Here, we took advantage of these key TMS attributes to map the spatiotemporal features of modulations affecting the motor system during SAT regulation in humans. A total of 50 participants performed a modified version of the "tokens task," involving choices between left and right index fingers. Incorrect choices led either to a high or to a low penalty in 2 different SAT contexts, inciting participants to emphasize either cautious or hasty decision policies, respectively. We tested 2 groups of participants in which TMS was applied at different stages of the decision-making task, either over the finger representations (TMS$_{Finger}$ participants; bilateral M1 TMS with a double-coil procedure, eliciting simultaneous MEPs in 3 finger muscles on both sides) or over the leg representation (TMS$_{Leg}$ participants; unilateral TMS with single-coil over left M1).

We focused on 2 main aspects of the MEP data. First, we considered the amplitude of MEPs from all finger and leg muscles over the course of the deliberation process in the 2 SAT contexts. This allowed us to assess the spatiotemporal features of motor excitability changes associated with SAT regulation in the task. Second, we considered the relationship between excitability changes shaping the chosen index finger and those occurring in the other finger muscles on the side of both the chosen and unchosen index fingers in the 2 contexts. To do so, we focused on data in the TMS$_{Finger}$ participants, where we obtained simultaneous MEPs from 6 muscles (3 on each side) in each trial. More precisely, we investigated the degree to which the trial-by-trial MEP variation in the chosen index finger related to the trial-by-trial MEP variation in the other finger muscles. The rationale here was that a high positive correlation between the chosen index and the other finger muscles would indicate the operation of influences exerting a broad, common impact on their motor representations [52–57], shaping MEPs in block. In contrast, a low or even a negative correlation would indicate the presence of influences affecting the chosen index representation in a more selective and differentiated way [52,54,58]. We compared the correlation values obtained in the hasty and the cautious contexts during deliberation.

Altogether, our data support the view that hastiness entails a broad amplification of motor excitability. As such, the hasty context was associated with particularly large MEPs, including in the leg muscles, although this effect was not entirely global as it was limited to the chosen side; it did not extend to muscles on the unchosen body side. Interestingly, on top of this effect, we also identified a local suppression of motor excitability, surrounding the index representation, also on the chosen side. Hence, a decision policy favoring speed over accuracy appears to rely on overlapping processes producing a broad (but not global) amplification and a surround suppression of motor excitability. The latter effect may help to increase the signal-to-noise ratio of the chosen representation, as also supported by the correlation analyses indicating a stronger differentiation of excitability changes between the chosen index and the other finger representations in the hasty relative to the cautious context.

## Results

On each trial of the tokens task, 15 tokens jump one by one every 200 ms from a central circle to 1 of 2 lateral target circles. Here, participants had to choose between left and right index finger key presses depending on which lateral circle they thought would ultimately receive the majority of the tokens (Fig 1A). They were free to respond at any time from Jump$_1$ to Jump$_{15}$. Correct and incorrect choices led to rewards and penalties, respectively (Fig 1B). The reward provided for correct choices decreased over the course of the trial, producing an increasing

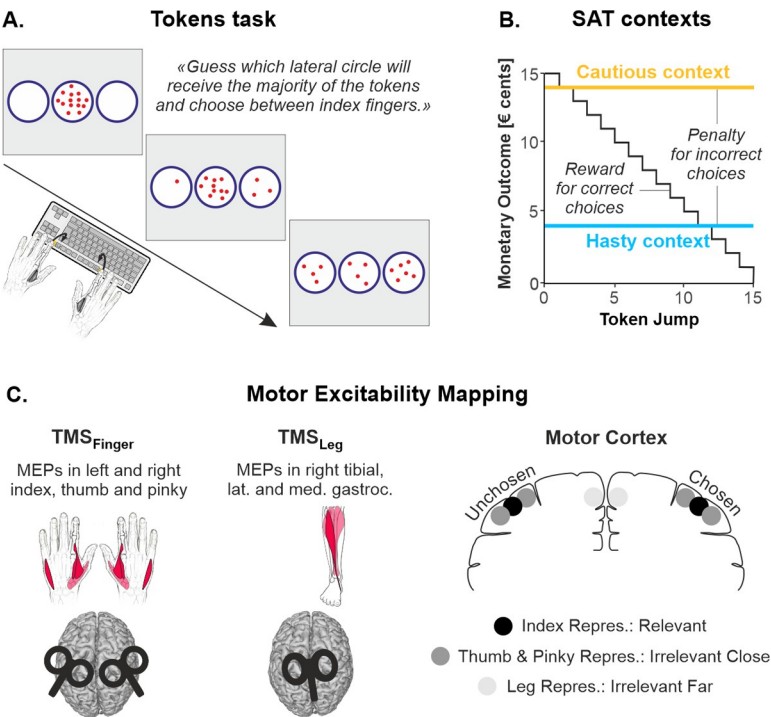

**Fig 1. Tokens task, SAT contexts, and motor excitability mapping. (A)** Tokens task. Participants had to choose between left or right index finger key presses depending on which lateral circle they thought would ultimately receive the majority of the tokens. **(B)** SAT contexts. The reward provided for correct choices decreased over the course of the trial, producing an increasing urge to decide. Most importantly, the use of a low penalty (−4 cents; blue) promoted hasty choices (hasty context), while a high penalty (−14 cents; yellow) fostered cautious choices (cautious context). In both contexts, a low penalty (−4 cents) was provided when participants did not respond before Jump$_{15}$ (not represented here). **(C)** Motor excitability mapping (related to S1 Fig). Left: In TMS$_{Finger}$ participants, a double-coil approach allowed us to elicit MEPs in index, thumb, and pinky muscles of both hands. In TMS$_{Leg}$ participants, MEPs were recorded in 3 right leg muscles. MEPs recorded in these 9 muscles were of reliable amplitude and were reproducible across sessions (see S1 Fig). Right: MEPs obtained in these different muscles allowed us to quantify excitability changes associated with a choice-relevant motor representation (i.e., the index finger representation), choice-irrelevant representations that lie close by the choice-relevant one (i.e., the thumb and pinky representations), and choice-irrelevant representations that lie far from the choice-relevant one (i.e., the leg representations). Further, classifying MEPs according to whether they fell on the same side as the chosen index or on the side of the unchosen index allowed us to measure excitability of the motor system on both the chosen and the unchosen side. MEP, motor-evoked potential; SAT, speed–accuracy trade-off; TMS, transcranial magnetic stimulation.

urge to decide. Most importantly, in one type of block, we sanctioned incorrect choices severely, with a penalty of −14 cents, emphasizing the need for cautiousness (cautious context). Conversely, the penalty provided for incorrect choices was only of −4 cents in the second block type, encouraging participants to make hasty decisions (hasty context).

We exploited single-pulse TMS over M1 in 2 subgroups of participants to quantify changes in motor excitability occurring in distinct representations, by probing MEPs in 9 different muscles (Fig 1C, S1 Fig). In TMS$_{Finger}$ participants ($n$ = 21), a double-coil approach was used to stimulate simultaneously the finger representations of both M1s. MEPs were recorded, concurrently in both hands, in an index, a thumb, and a pinky muscle. The index muscle being the prime mover in the task, its MEPs allowed us to quantify motor excitability changes in a choice-relevant motor representation. The thumb and pinky muscles being not required in the task, their MEPs allowed us to assess excitability changes associated with choice-irrelevant representations that lie close by the prime mover representation in the motor system (i.e., in terms of somatotopy). In TMS$_{Leg}$ participants ($n$ = 22), we stimulated the left leg

representation and recorded MEPs in 3 right leg muscles. These muscles being not required in the task, their MEPs allowed us to assess changes associated with choice-irrelevant representations that lie far from the prime mover in terms of somatotopy. Further, because the task required deciding between right and left index finger choices, MEPs could be classified according to whether they fell on the same side as the chosen index or on the side of the unchosen index, providing us with measures of excitability for each side in both the $TMS_{Finger}$ and the $TMS_{Leg}$ participants. Finally, a subgroup of No-TMS participants ($n = 7$) performed the task without stimulation, allowing us to control for any effect of TMS on decision behavior.

## Participants regulated their decision behavior depending on context

To highlight the presence of a SAT in our task, we regressed individuals' percentages of correct choices (i.e., accuracy) against their decision times (DTs) using a permutation-based correlation ($N_{Participants} = 50$, $N_{Permutations} = 1,000$). As expected, this analysis showed a significant positive correlation between DTs and accuracy, with participants presenting the fastest DTs being also the least accurate, both in the hasty and in the cautious contexts ($R = 0.72$ and $R = 0.68$, respectively, both $p$-values $< 0.0001$; Fig 2A). Most importantly, the participants' distribution appeared shifted in the hasty relative to the cautious context, supporting a change in SAT (e.g., see distributions in the margins of Fig 2A). Indeed, a between-context comparison revealed that both DTs and accuracy were significantly lower in the hasty context ($t_{49} = -8.42$, $p < 0.0001$, Cohen's d = 1.191 and $t_{49} = -11.26$, $p < 0.0001$, Cohen's d = 1.593, respectively;

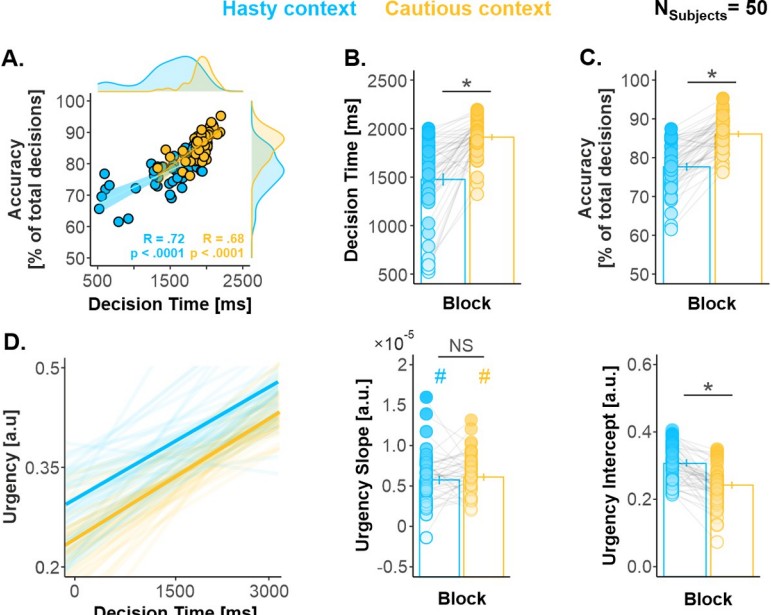

**Fig 2. (related to S1 Table and S2 Fig). Participants shifted their SAT in the hasty relative to the cautious context.** (A) Speed–accuracy relationship. A permutation-based correlation ($N_{Permutations} = 1,000$) revealed a significant positive correlation between individuals' DTs and accuracy, present in both contexts. As expected, the fastest participants were also the least accurate, highlighting the presence of a SAT in the task. (B) DTs. Participants presented significantly faster DTs in the hasty compared to the cautious context. (C) Decision accuracy. Accuracy was significantly lower in the hasty context. (D) Urgency functions. While the slope of the functions was comparable in the hasty and the cautious contexts (middle panel), the intercept was significantly higher in the former context. As a result, the level of urgency was higher in the hasty context throughout the deliberation period. *: significant effect of context at $p < 0.05$. Error bars represent 1 SEM. All individual and group-averaged numerical data exploited for Fig 2 are freely available at this link: https://osf.io/tbw7h. DT, decision time; SAT, speed–accuracy trade-off.

Fig 2B and 2C). Altogether, these findings show that participants regulated their SAT in accordance with our expectations, favoring a hasty policy when the context involved a low penalty and emphasizing cautiousness when a high penalty was at stake.

Next, we tested whether participants exhibited changes in the level of urgency from one context to another. To do so, we extracted urgency functions using a previously described computational analysis of decision behavior [59]. As predicted by time-varying models of decision-making [1,43,60,61], we found that urgency increased significantly as time elapsed during deliberation, both in the hasty and in the cautious contexts (i.e., $t$ tests against 0 on slope values: $t_{49} = 14.58$, Cohen's d = 2.069 and $t_{49} = 19.0$, Cohen's d = 2.667, respectively, Bonferroni-corrected $p$-values < 0.0001; Fig 2D). Most importantly, while the slope of the functions did not differ significantly between contexts ($t_{49} = -0.82$, $p = 0.419$, Cohen's d = 0.124), the intercept was significantly higher in the hasty than in the cautious context ($t_{49} = 7.42$, $p < 0.0001$, Cohen's d = 1.050). Together, these findings indicate that the level of urgency was higher in the hasty than in the cautious context at the start of the deliberation period and that this difference persisted throughout that period.

The experimental plan entailed counterbalancing the session order between participants. Yet, this balancing was not perfect and out of the 50 participants included in the behavioral analysis, 24 participants started the experiment with the hasty session, while 26 began with the cautious one. To ensure that the effects of context observed on DT, accuracy and urgency intercept did not depend on this imperfect counterbalancing, we performed Bayesian ANOVAs, testing whether the factor CONTEXT interacted with SESSION ORDER (see S1 Table). Bayes factors (BFs) ranged between 3.22 and 4.45, providing strong evidence for a lack of CONTEXT*SESSION ORDER interaction on decision behavior, indicating that none of the effects of CONTEXT reported above depended on the SESSION ORDER.

In addition, we investigated the potential impact of TMS on behavior. To do so, we analyzed the DT, accuracy, as well as the slope and intercept of the urgency functions while considering the TMS subgroup (i.e., TMS_Finger, TMS_Leg, No-TMS participants) as a categorical predictor in our analyses (i.e., using ANOVAs). We did not find any significant effect of the TMS subgroup on the behavioral data, nor of its interaction with the factor CONTEXT (i.e., hasty versus cautious context; S2 Fig). Further, a BF analysis provided evidence for a lack of effect of the subgroup on all of these behavioral data (all BFs ranged between 4.06 and 8.35). In fact, the 3 subgroups presented very similar effects of context on decision behavior. This indicates that the application of TMS over the finger or leg representations did not perturb SAT regulation in this task.

## Motor excitability globally increased as time elapsed during deliberation

To assess the dynamics of motor excitability changes over the decision period, we applied TMS in 90% of trials, at 1 of 4 possible timings during the task: at Jump$_1$, Jump$_4$, and Jump$_7$, as well as at baseline (i.e., between the trials; Fig 3A). In order to capture excitability changes related to deliberation, we selected trials in which responses occurred at least 150 ms after Jump$_7$ and up to Jump$_{15}$. Further, to prevent MEP amplitudes from being affected by the difference in decision speed between each context, we homogenized the reaction time (RT) distributions across contexts by selecting trials through a RT-matching procedure, a procedure previously exploited in the SAT literature [1] (S3 Fig). Following the RT-matching procedure, we had to exclude 2 out of the 21 TMS_Finger participants and 6 out of the 22 TMS_Leg participants, as they ended up with less than 8 trials on average per timing and context [62]. Hence, 35 participants were considered for the MEP analyses (19 TMS_Finger and 16 TMS_Leg participants). Note that the behavioral effects reported in the previous section, including the presence

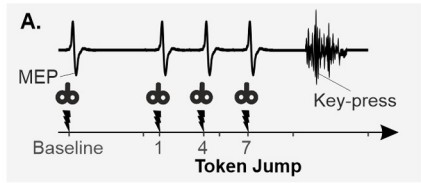

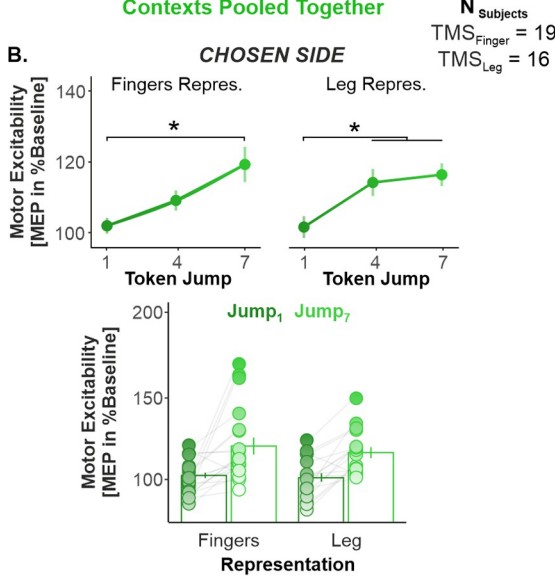

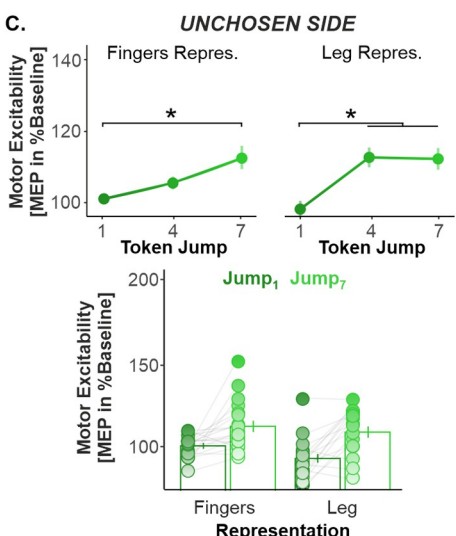

**Fig 3. Motor excitability globally increased as time elapsed during deliberation.** **(A)** TMS timings. We applied TMS at 4 timings during the task: at baseline (i.e., between the trials) and at $Jump_1$, $Jump_4$, and $Jump_7$ during the decision period. **(B)** Main effect of TIMING on motor excitability on the chosen side. To highlight the main effect of TIMING, both SAT contexts were pooled together. Further, the data obtained for the index, the thumb, and the pinky representations (top left) were averaged for the figure as well as the data obtained for the 3 leg representations (top right). The bar graph at the bottom displays the individual data points, as obtained at $Jump_1$ (dark green) versus $Jump_7$ (light green). Overall, this representation shows that a large proportion of participants exhibited an increase in motor excitability from $Jump_1$ to $Jump_7$, both in the finger and in the leg representations. **(C)** Same as B. for the unchosen side. Error bars represent 1 SEM. * significant effect of timing at $p < 0.05$. All individual and group-averaged numerical data exploited for Fig 3 are freely available at this link: https://osf.io/tbw7h. SAT, speed–accuracy trade-off; TMS, transcranial magnetic stimulation.

of a SAT shift, were still significant when considering this smaller group of participants (see S4 Fig). In order to investigate the motor correlates of the SAT shift, we normalized MEP amplitudes obtained at $Jump_1$, $Jump_4$, and $Jump_7$ as a percentage of baseline [63] for each motor representation and for both the chosen and the unchosen sides, in the 19 $TMS_{Finger}$ and 16 $TMS_{Leg}$ participants.

A 3-way repeated measures [rm]ANOVA with CONTEXT, TIMING, and REPRESENTATION as within-participants factors revealed that normalized MEP amplitudes displayed a main effect of TIMING in $TMS_{Finger}$ participants (i.e., $Jump_1$ versus $Jump_4$ versus $Jump_7$). This effect was significant for both the chosen (Greenhouse–Geiser [GG] corrected: $F_{1.4,25.4} = 10.610$, $p = 0.001$, partial $\eta^2 = 0.371$; Fig 3A) and the unchosen side MEPs (GG corrected: $F_{1.5,26.3} = 8.3024$, $p = 0.003$, partial $\eta^2 = 0.283$; Fig 3B). Tukey honestly significant difference (HSD) post hoc tests (corrected for 3 between-pair comparisons) revealed that amplitudes were significantly larger at $Jump_7$ compared to $Jump_1$ and $Jump_4$ (all $p$-values = [0.0002, 0.0473]). Surprisingly, a similar effect of TIMING was observed in $TMS_{Leg}$ participants, with MEPs growing over time on both the chosen ($F_{2,30} = 10.206$, $p = 0.0007$, partial $\eta^2 = 0.405$; Fig 3A) and the unchosen side ($F_{2,30} = 21.716$, $p < 0.0001$, partial $\eta^2 = 0.591$; Fig 3B). Here, post hoc tests revealed that amplitudes were larger at $Jump_4$ and $Jump_7$ than at $Jump_1$ (all $p$-values = [0.0012, 0.0001]). Overall, these findings indicate that motor excitability exhibited a global increase as time elapsed during the decision process. This time-dependent effect not only concerned choice-relevant representations, but also choice-irrelevant ones, and even those lying far away within the motor system (leg), on both the chosen and the unchosen sides.

## Hastiness relied on a broad amplification and a surround suppression of motor excitability on the chosen side during deliberation

Importantly, MEP amplitudes also showed a significant effect of CONTEXT on the chosen side, which varied as a function of the TIMING and of the finger REPRESENTATION in $TMS_{Finger}$ participants (CONTEXT*TIMING*REPRESENTATION interaction: $F_{4,72} = 3.63$, $p = 0.009$, partial $\eta^2 = 0.168$; see Fig 4A). Lower-level significant effects included a TIMING*REPRESENTATION ($F_{4,72} = 3.01$, $p = 0.023$, partial $\eta^2 = 0.143$) and a CONTEXT*REPRESENTATION interaction ($F_{2,36} = 3.93$, $p = 0.028$, partial $\eta^2 = 0.179$); however, we focus hereafter on the higher-level CONTEXT*TIMING*REPRESENTATION interaction for the purpose of narrative clarity. Tukey HSD post hoc tests performed following the CONTEXT*TIMING*REPRESENTATION interaction (corrected for 153 between-pair comparisons) revealed that the MEPs obtained in the index muscle were significantly larger in the hasty than in the cautious context at $Jump_7$ ($p = 0.021$), indicating an amplification of motor excitability in the choice-relevant representation. Notably, $TMS_{Leg}$ participants also displayed a significant main effect of CONTEXT on the chosen side ($F_{1,15} = 4.65$, $p = 0.047$, partial $\eta^2 = 0.237$; see Fig 4A, right panel), with leg MEPs being larger in the hasty than in the cautious context, indicating that the amplification of motor excitability also affected the leg representations. This effect was reproducible across the 3 investigated leg muscles (nonsignificant GG-corrected CONTEXT*TIMING*REPRESENTATION interaction $F_{2.4,36.6} = 1.54$, $p = 0.226$, partial $\eta^2 = 0.093$; see S5 Fig). Hence, on the chosen side, MEPs were significantly larger in the hasty than in the cautious context, and this effect not only concerned the choice-relevant (chosen) muscle, but also choice-irrelevant leg muscles that lie far from the prime mover. Besides that, post hoc tests performed on the $TMS_{Finger}$ participants' data (i.e., following the CONTEXT*TIMING*REPRESENTATION interaction mentioned above) also revealed an additional effect that concerned specifically the thumb and pinky muscles on the chosen side. Here, MEPs were significantly smaller in the hasty than in the cautious context at $Jump_7$

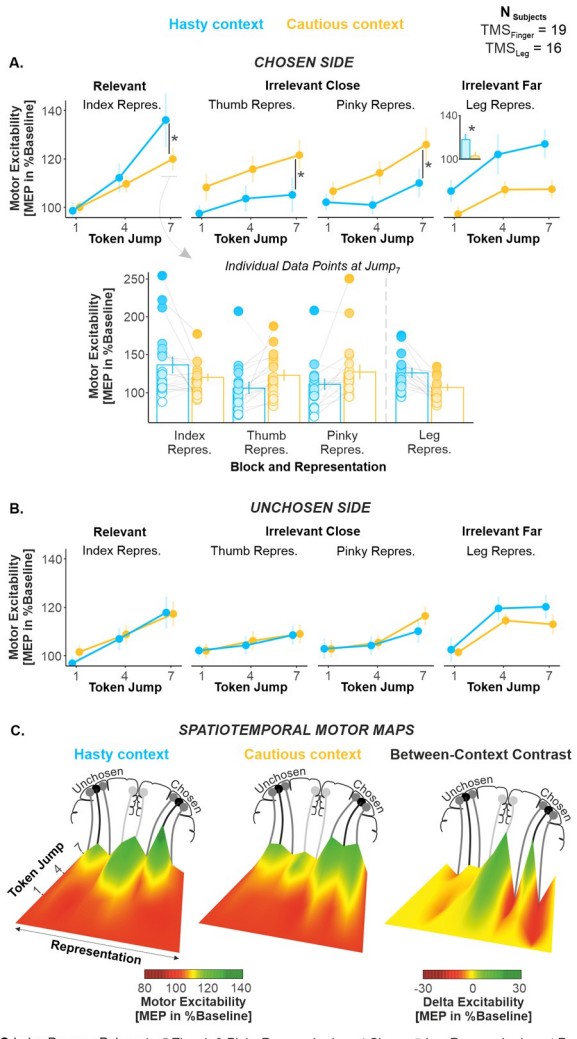

**Fig 4. (related to S4–S6 Figs as well as S2 and S3 Tables). Hastiness relied on a broad amplification and a surround suppression of motor excitability on the chosen side. (A)** Effect of CONTEXT on motor excitability on the chosen side. The top graphs show excitability changes occurring over the decision period in the hasty and cautious contexts (blue and yellow traces, respectively). Given that the effect of context was reproducible across the 3 leg representations (see S5 Fig), the MEP data have been pooled together (right panel). The inset in the right panel denotes the main effect of context on the leg region. The bar graph at the bottom displays individual data points as obtained at Jump7, for each context and each representation. Error bars represent 1 SEM.*: significant effect of context at $p < 0.05$. **(B)** Same as A. for the unchosen side. **(C)** Spatiotemporal motor maps. We computed spatiotemporal maps to provide an integrative view of motor excitability changes occurring during the course of deliberation in each context (see main text). To this aim, we considered altogether the MEPs obtained for the index, thumb, pinky, and leg representations of the chosen and unchosen sides, and we arranged them spatially according to M1 somatotopy. One spatiotemporal map was obtained for each context, and a between-context difference map was finally computed (i.e., hasty minus cautious context). The difference map (right panel) highlights the increase in excitability in the index and leg representations of the chosen side (right side of the map, green, positive values) as well as the surround suppression occurring in the thumb (more lateral) and pinky (more medial) representations (red, negative values). Besides, no noticeable between-context difference emerged on the unchosen side (left side of the map, yellow values). All individual and group-averaged numerical data exploited for Fig 4 are freely available at this link: https://osf.io/tbw7h. M1, primary motor cortex; MEP, motor-evoked potential; TMS, transcranial magnetic stimulation.

($p = 0.013$ and 0.0247 for the thumb and pinky fingers, respectively; but see S2 Table for an alternative post hoc analysis), suggesting thus a suppression of excitability in the surrounding choice-irrelevant representations. Altogether, these data indicate that the SAT shift observed

in the hasty context was associated with the occurrence of 2 overlapping modulatory changes on the chosen side: a broad amplification expanding to remote choice-irrelevant representations and a local suppression of choice-irrelevant representations surrounding the choice-relevant one.

As for the behavioral analysis, the session order was not completely counterbalanced among the 19 TMS$_{Finger}$ participants included in the MEP analysis: 8 participants started the experiment with the hasty session and 11 with the cautious one. Besides, the session order was counterbalanced in the 16 TMS$_{Leg}$ participants (i.e., 8 started with the hasty session). Here, too, we controlled for the imperfect balancing of the session order by performing Bayesian ANOVAs, testing whether the factor CONTEXT interacted with SESSION ORDER (see S3 Table). All BFs ranged between 3.08 and 30.09, providing strong to decisive evidence for a lack of effect of the session order on the effect of context on motor excitability.

Interestingly, these modulatory changes did not involve the unchosen side. As evident in Fig 4B, MEP amplitudes obtained there did not show any significant effect of CONTEXT ($F_{1,18} = 0.57$, $p = 0.457$, partial $\eta^2 = 0.031$ and $F_{1,15} = 1.06$, $p = 0.318$, partial $\eta^2 = 0.066$ for TMS$_{Finger}$ and TMS$_{Leg}$ participants, respectively), nor did they show any CONTEXT*TIMING ($F_{2,36} = 0.01$, $p = 0.987$, partial $\eta^2 = 6.88 \times 10^{-5}$ and $F_{2,30} = 0.65$, $p = 0.529$, partial $\eta^2 = 0.041$), CONTEXT*REPRESENTATION ($F_{2,36} = 0.06$, $p = 0.935$, partial $\eta^2 = 0.004$ and $F_{2,30} = 0.589$, $p = 0.561$, partial $\eta^2 = 0.037$) or CONTEXT*TIMING*REPRESENTATION interaction ($F_{4,72} = 1.22$, $p = 0.310$, partial $\eta^2 = 0.063$ and $F_{4,60} = 0.47$, $p = 0.756$, partial $\eta^2 = 0.030$). BFs for all of these effects ranged between 7.82 and 492.78, providing further evidence for a lack of effect of context on the unchosen side [64]. Hence, while the chosen side undergoes a broad amplification adding up to a local suppression when decisions have to be fast, representations on the unchosen side remain largely unaffected by the context.

Importantly, the RT-matching procedure described above ensured similar RTs between the 2 contexts, but it raises a potential confound by emphasizing the slowest trials from the hasty context and the fastest trials from the cautious context. However, concerns about that confound are reduced by the observation that the same analyses performed on the full set of trials, without RT matching, produced the same results (see S6 Fig).

Based on these data, we computed spatiotemporal maps to provide an integrative view of motor excitability changes occurring during the course of deliberation (i.e., for each stimulation time) in each context (Fig 4C). To this aim, we considered altogether the MEPs obtained for the index, thumb, pinky and leg representations on the side of both the chosen and unchosen index fingers (i.e., 8 representations), in each context. The 8 traces were spatially arranged according to M1 somatotopy (i.e., from lateral to medial: thumb, index, pinky, and leg), and a linear interpolation was performed to estimate excitability changes between each representation. Two spatiotemporal maps were obtained (one for each context), and a between-context contrast map was finally computed (i.e., hasty minus cautious context). The difference map highlights the increase in excitability (in green) for the index of the chosen side, expanding to leg representations, as well as the suppression (in red) occurring in the surrounding thumb and pinky representations (Fig 4C, right panel).

## Hastiness did not affect baseline activity

The data presented in Fig 4 highlight the effects of context on motor excitability during deliberation. Next, we wanted to assess whether context also altered excitability outside the deliberation period, when participants were resting between trials. To do so, MEP amplitudes obtained at the baseline timing (see Fig 3A) were normalized with respect to MEPs recorded at rest, outside of the task [63]. We did not apply any RT-matching procedure on these data, as

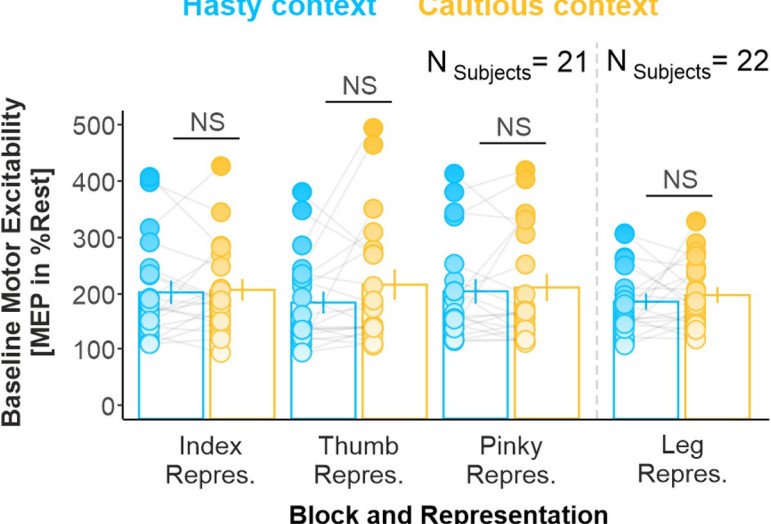

**Fig 5. Hastiness did not affect baseline activity.** NS denotes the lack of significant difference between both contexts. Error bars represent 1 SEM. All individual and group-averaged numerical data exploited for Fig 5 are freely available at this link: https://osf.io/tbw7h. MEP, motor-evoked potential.

the baseline timing was deemed too far from the deliberation period to be affected by any between-context difference in decision speed. This allowed us to include every participant in the analysis (i.e., 21 TMS$_{Finger}$ participants and 22 TMS$_{Leg}$ participants).

Interestingly, we did not find any significant effect of the factor CONTEXT (TMS$_{Finger}$ participants: $F_{1,20} = 1.14$, $p = 0.297$, partial $\eta^2 = 0.054$; TMS$_{Leg}$ participants: $F_{1,21} = 0.59$, $p = 0.451$, partial $\eta^2 = 0.027$) nor was there any interaction with the factor REPRESENTATION (TMS$_{Finger}$ participants: $F_{2,40} = 1.49$, $p = 0.236$, partial $\eta^2 = 0.069$; TMS$_{Leg}$ participants: $F_{2,42} = 2.17$, $p = 0.127$, partial $\eta^2 = 0.093$) on baseline MEPs (Fig 5). Further, a BF analysis provided substantial evidence for a lack of effect of CONTEXT on these data (BFs = 4.41 and 3.55, for TMS$_{Finger}$ and TMS$_{Leg}$ participants, respectively) and of the CONTEXT*REPRESENTATION interaction (BFs = 9.53 and 4.93). Altogether, these results indicate that the effect of context on motor excitability was restricted to the deliberation period, leaving baseline activity unaffected.

### Hastiness was associated with a decorrelation between the chosen index and the other finger representations during deliberation

To further characterize the impact of context at Jump$_7$ (Fig 4A), we quantified, in each context, the relationship between trial-by-trial MEP variations in the chosen index and in the 5 other fingers (Fig 6A). We considered the trials of all participants (N$_{Participants}$ = 19), providing us with a large pool of data points (N$_{Trials}$ = 528) and adopted a repeated measures correlation (rmCorr) [65] to estimate statistical significance of each of the 10 correlations (Bonferroni-corrected significance threshold at $p = 0.005$). RmCorr accounts for nonindependence among observations using analysis of covariance (ANCOVA) to statistically adjust for interindividual variability. By removing measured variance between participants, rmCorr provides the best linear fit for each participant using parallel regression lines (the same slope) with varying intercepts. Hence, rmCorr tends to have much greater statistical power relative to simple correlation analysis because neither averaging nor aggregation is necessary for an intraindividual research question [65]. Once the correlation coefficient obtained, for each of the 5 pairs of muscles, we compared the strength of the correlation in the hasty relative to the cautious

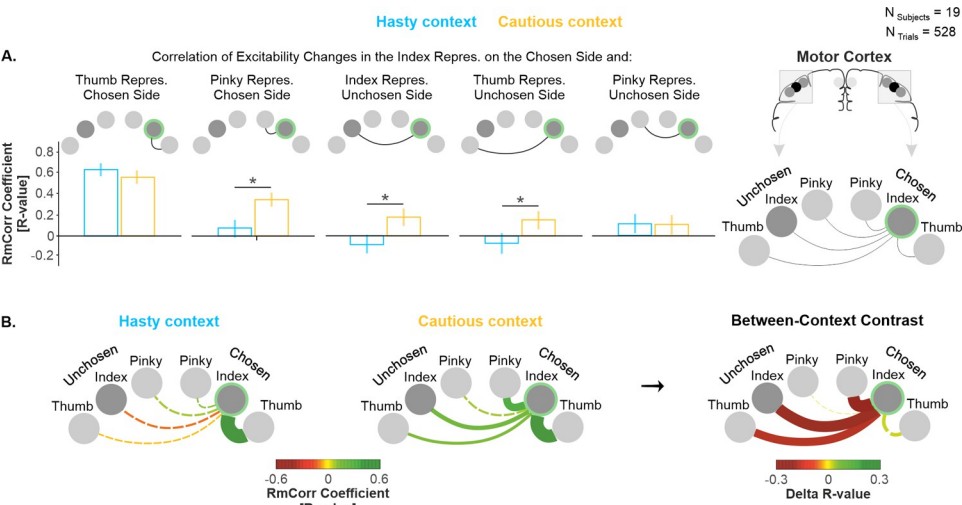

**Fig 6.** (related to S7 Fig). **Hastiness was associated with a decorrelation of excitability changes between the chosen index and the other finger representations during deliberation. (A)** R-values obtained from the rmCorr analysis. Error bars represent 95% CI. * indicates a significant difference between R-values at *p* < 0.01 (Bonferroni-corrected threshold), detected using Fisher Z test. **(B)** Network plots. The color and thickness of the lines in the left and middle panels represent the strength of the correlations, as indexed by RmCorr's R-values. Solid and dashed lines represent significant and nonsignificant correlations, respectively. In the right panel, the color and thickness of the lines represent the difference in R-values between the cautious and the hasty context. Solid and dashed lines represent significant and nonsignificant different in correlation, respectively, as determined using Fisher Z test. All individual and group-averaged numerical data exploited for Fig 6 are freely available at this link: https://osf.io/tbw7h. CI, confidence interval; rmCorr, repeated measures correlation.

context based on the 95% confidence intervals (CIs) calculated for each R-value using the rmCorr procedure.

Interestingly, MEPs of the chosen index correlated positively with MEPs of all other fingers in the cautious context, whether on the chosen or on the unchosen side (i.e., all of the 5 R-values were positive; Fig 6A, S7 Fig). Further, 4 of the 5 correlations were significant in the cautious context (*p*-values = [0.000001, 0.003]). Conversely, MEPs of the chosen index correlated positively with MEPs of only 3 other fingers in the hasty context (i.e., 3 R-values were positive, and the 2 others were negative), and only 1 of the positive correlations was significant (*p* < 0.00001). Hence, the rmCorr analysis suggests a weaker trial-by-trial positive relationship in excitability changes between the chosen index and the other fingers in the hasty context. Consistently, 95% CIs calculated for these R-values revealed that the R-value was often significantly weaker in the hasty than in the cautious context (see Fig 6A). Indeed, 95% CIs did not overlap between contexts when considering the link on the chosen side with the pinky (R = 0.08 versus R = 0.34 in the hasty and cautious contexts, respectively, and CIs = [−0.007, 0.16] versus [0.27, 0.42]) and on the unchosen side with the index (R = −0.08 versus R = 0.17 and CIs = [−0.17, 0.005] versus [0.08, 0.25], respectively) and the thumb (R = −0.06 versus R = 0.13 and CIs = [−0.14, 0.03] versus [0.04 0.21]). Fig 6B provides a visual synthesis of these effects. Altogether, these data indicate that hastiness involves a decoupling of excitability in the chosen finger representation with respect to the other finger representations, which may help enhancing the signal-to-noise ratio of the chosen representation [53].

## Movement vigor was unaffected by elapsed time and hastiness

Given the known impact of movement vigor on motor activity [3,66,67], we investigated electromyography (EMG) activity to test whether any change in vigor could have contributed to

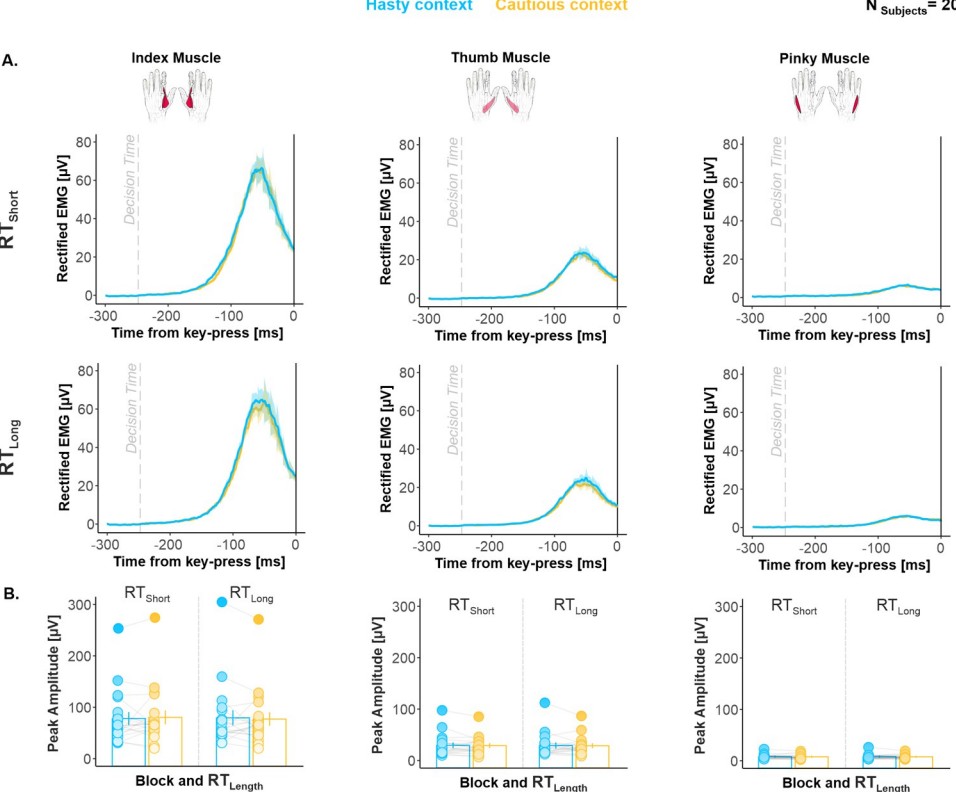

**Fig 7. (related to S8 Fig). Movement vigor was unaffected by elapsed time and hastiness. (A)** Group-averaged rectified EMG activity. Shaded error bars represent 1 SEM. The vertical dotted line indicates the estimated DT (see Materials and methods). **(B)** Group-averaged peak amplitude. Error bars represent 1 SEM. Overall, EMG activity was comparable for $RT_{Short}$ and $RT_{Long}$ as well as across contexts in each of the 3 muscles. All individual and group-averaged numerical data exploited for Fig 7 are freely available at this link: https://osf.io/tbw7h. DT, decision time; EMG, electromyography; RT, reaction time.

the changes in MEP amplitude observed in our task. To this aim, we exploited the EMG signals recorded in the moving hand of $TMS_{Finger}$ participants (i.e., in the index, thumb, and pinky muscles) and considered the voluntary contraction preceding the key press in the 2 contexts. For each response provided, we rectified the signal and extracted the peak amplitude as a proxy of movement vigor [2,68]. To investigate the effect of elapsed time on this variable in each context, we split the trials into 2 subsets according to whether they were associated with short or long RTs, using a median-split procedure ($RT_{Short}$ and $RT_{Long}$ trials, respectively; S8 Fig). Further, to prevent EMG peak amplitude from being affected by the difference in decision speed between each context, we homogenized the RT distributions across contexts through a RT-matching procedure [1], both for $RT_{Short}$ and $RT_{Long}$ trials. Following this procedure, we had to exclude 1 out of the 21 $TMS_{Finger}$ participants, as he/she ended up with no trial in a specific condition ($N_{Participants}$ = 20).

As evident in Fig 7 (see also S8 Fig), the analysis of EMG peak amplitude did not show any significant effect of CONTEXT ($F_{1,19}$ = 0.007, $p$ = 0.934, partial $\eta^2$ = 0.003) or $RT_{LENGTH}$ ($F_{1,19}$ = 0.09, $p$ = 0.763, partial $\eta^2$ = 0.004). There was also no CONTEXT*$RT_{LENGTH}$ ($F_{2,36}$ = 0.86, $p$ = 0.365, partial $\eta^2$ = 0.043) or CONTEXT*$RT_{LENGTH}$*MUSCLE interaction (GG corrected: $F_{1.1,20.1}$ = 1.81, $p$ = 0.193, partial $\eta^2$ = 0.087). BFs for all of these effects ranged between 7.07 and 7.32, providing evidence for a lack of effect of elapsed time and context on EMG peak

amplitude. Hence, the time- and context-dependent changes in motor excitability observed in our task cannot be accounted for by variations in movement vigor.

## Discussion

The goal of the present study was to test the hypothesis that SAT regulation relies on a global modulation of motor activity during sensorimotor decisions [1–3]. Participants performed a task involving choices between left and right index fingers, in which incorrect choices led either to a high or to a low penalty in 2 contexts, inciting them to emphasize either cautious or hasty decision policies, respectively. We applied TMS on different motor representations in M1, eliciting MEPs in multiple finger and leg muscles at different stages of the decision-making task. MEP amplitudes allowed us to probe activity changes in the corresponding finger and leg representations, while participants were deliberating about which index finger to choose.

### Participants regulated their decision behavior depending on context

Overall, participants made faster but less accurate choices when the context involved a low penalty, relative to when the penalty was high. Furthermore, a computational analysis of the behavioral data revealed that participants exhibited a higher level of urgency in the low penalty context. The latter analysis also showed that urgency grew as time elapsed during the deliberation period, replicating previous findings in the literature [1,43,60,61]. Altogether, our decision data indicate that reducing the cost of incorrect choices increased participants' urge to act, leading them to shift their SAT from a cautious to a hasty decision policy.

### Hastiness relied on a broad amplification of motor excitability on the chosen side during deliberation

The SAT shift observed in the hasty context was associated with a broad amplification of excitability on the chosen side during deliberation, altering both the choice-relevant (index) representation and remote choice-irrelevant (leg) representations. Importantly, though, this modulation did not globally impact the motor system, leaving the unchosen side unaffected. Hence, these results do not support the idea of a global modulation of motor activity across contexts. Rather, they suggest the existence of broad neural sources pushing up motor activity unilaterally when context calls for hasty decisions, ensuring a faster rise-to-threshold of neural activity in the chosen hemisphere and thus biasing the competition between motor representations. Given their strong ipsilateral projections to the motor cortex [69] and their known involvement in SAT regulation [25,28–30], the basal ganglia represent a potential candidate for this unilateral, broad amplification, a hypothesis worthy of further investigation.

### Hastiness did not affect baseline activity

Interestingly, the broad amplification was restricted to the deliberation period and did not affect baseline activity. This result may appear to be at odds with previous findings of the literature, showing upward shifts in baseline activity in hasty contexts [3,4]. However, these so-called baseline shifts are most often observed right before the decision period [3,26], while baseline measures were probed long before the start of the decision period in the present study (i.e., 1,300 ms before the first token jump). As such, the impact of context on baseline activity may depend on the state of motor preparation, becoming stronger as the decision period—and therefore the need to act—draws nearer. In line with this interpretation, motor excitability was already amplified at the beginning of the decision period (i.e., at $Jump_1$) in the leg representations. Alternatively, it is possible that a fraction of corticospinal cells showed an amplification

of activity at our baseline timing, but that this effect canceled out at the population level, when probed with TMS. Indeed, baseline shifts are usually only observed for a fraction of neurons in single-cell studies [3,26]. At the population level, several studies in humans failed to observe such shifts in the motor cortex, whether using functional magnetic resonance imaging (fMRI) [29], TMS [23], or electroencephalogram (EEG) [1] (but see [2]).

## Hastiness relied a surround suppression of motor excitability on the chosen side during deliberation

In addition to relying on a broad amplification, hastiness was associated with a local suppression of motor excitability during deliberation, which affected choice-irrelevant representations surrounding the choice-relevant population on the chosen side. This effect is reminiscent of center-surround inhibition mechanisms [70–73], classically associated with lateral inhibition in the motor cortex. In fact, amplification of activity within the neurons of the choice-relevant representation may have resulted in an increased recruitment of inhibitory interneurons connecting them to adjacent populations. Such a mechanism may be of particular importance in the motor system, where populations of corticospinal cells projecting to different muscles may be recruited concurrently during action execution. Here, lateral inhibition could enhance the signal-to-noise ratio within the representations of the moving effector when context calls for hastiness, allowing for excitatory inputs targeting these representations to better stand out against a quiescent background [74–77], ultimately reducing the time needed to initiate the action following commitment.

## Hastiness was associated with a decorrelation of the chosen index and the other finger representations during deliberation

The correlation analyses also support the idea of an enhancement in signal-to-noise ratio within the chosen index representation in the hasty context. The rationale for these analyses was that a high positive correlation between MEPs obtained in the index and in the other finger muscles would indicate the operation of influences exerting a common impact on their representations [52,53], shaping MEPs in block. As such, shared neural inputs are known to produce correlated fluctuations of neural activity across functionally divergent populations [52–57]. Along these lines, previous studies have found significant correlations between neurons in diverse cortical areas [52–54,78–81]. Here, we found that excitability changes in the chosen index representation and in the other finger representations decorrelated in the hasty relative to the cautious context, possibly indicating the presence of influences affecting the chosen index representation in a more selective and differentiated way when hastiness was at premium [52,54,58]. In the visual cortex, a similar decorrelation of neural activity has been observed when attention is directed to a stimulus inside a population's receptive field [52,53]. Computational analyses revealed that this attention-driven decorrelation enhances the signal-to-noise ratio of pooled neural signals substantially [53], a finding in accordance with our current interpretation.

## Motor excitability globally increased as time elapsed during deliberation

Beyond these context-dependent effects, our data also unveiled an interesting effect of time on motor excitability. In fact, motor excitability displayed a global rise over time during the decision period, which affected all of the representations investigated in the present study. Previous work has shown that activity often rises concomitantly in different choice-relevant representations during sensorimotor decisions (e.g., [13,15,82]). This finding is usually considered to

reflect the unfolding of a competition between neural populations involved in the decision process [19,20,83]. However, our data show that neural activity also builds up over time in choice-irrelevant representations. One potential explanation for this new result is that diffuse modulatory inputs may progressively amplify activity in the sensorimotor system as the urge to act increases during deliberation. In line with this interpretation, several sensorimotor regions—such as the premotor cortex [84,85], the lateral intraparietal area [86,87], or the cerebellum [18,88,89]—display time-dependent ramping activities during decision-making. Given its diffuse projections to these structures [90], the noradrenergic system may represent a potential candidate for this time-dependent modulation. In support of this hypothesis, pupil dilation—a proxy of noradrenergic activity [91]—also rises as time elapses during deliberation [2].

### Movement vigor was unaffected by elapsed time and hastiness

Our analysis of voluntary EMG activity suggests that none of these context- and time-dependent changes in motor excitability could be accounted for by alterations in movement vigor. Indeed, movement vigor was comparable in the hasty and cautious context as well as for short and long RTs. At first glance, this finding may appear to contrast with sensorimotor theories of decision-making, postulating that a common decision urgency/movement vigor mechanism would regulate decision and movement speeds [2,68,92–94]. However, recent studies have come to question this unified mechanism view, showing that decision and movement speeds do not necessarily covary systematically [92,95,96]. Our findings are therefore in line with these recent observations and suggest the putative contribution of distinct (yet, overlapping) neural sources to the invigoration of decision-making and action execution processes.

### Conclusions

Altogether, our data reveal the concurrent operation of multiple modulatory influences on the motor system during hasty sensorimotor decisions. We found that motor excitability exhibits a global increase as time elapses during the decision process, altering not only choice-relevant representations, but also choice-irrelevant ones that lie far away within the motor system, on both the chosen and the unchosen sides. Beyond this time-dependent effect, the data show that shifting from a cautious to a hasty context entails a broad amplification of motor excitability, although this amplification was not entirely global as it was limited to the chosen side. Interestingly, on top of this effect, we also identified a local suppression of motor excitability, surrounding the index representation on the chosen side. Hence, a decision policy favoring speed over accuracy appears to rely on overlapping processes producing a broad (but not global) amplification and a surround suppression of motor excitability. The latter effect may help increasing the signal-to-noise ratio of the chosen representation, as supported by the correlation analyses indicating a stronger decoupling of excitability changes between the chosen index representation and the other finger representations in the hasty relative to the cautious context.

### Materials and methods

#### Resource availability

**Lead contact.**   Further information and requests for resources can be directed to and will be fulfilled by the corresponding author, Gerard Derosiere: gerard.derosiere@uclouvain.be.
**Materials availability.**   This study did not generate new unique reagents.

**Data and code availability.**   All datasets and codes generated during this study will be freely available on the Open Science Framework repository upon publication at https://osf.io/tbw7h.

## Experimental model and participant details

**Participants.**   A total of 50 healthy human participants engaged in the study. Among them, 21 received TMS over the finger motor representations (i.e., $TMS_{Finger}$ participants; 11 women, 24 ± 0.5 years), and 22 received TMS over the leg representations (i.e., $TMS_{Leg}$ participants; 14 women, 22.7 ± 0.3 years); 7 did not receive TMS and were thus only considered for the behavioral analyses (i.e., No-TMS participants; 4 women, 21.7 ± 0.5 years [mean ± SE]). The latter participants were the first ones we tested with the goal to verify that the different penalty induced a shift in SAT between contexts and that we could thus implement TMS in it. $TMS_{Finger}$ and $TMS_{Leg}$ participants were then included with the objective to reach a sample of 50 participants in total.

Participants who received TMS answered a medical questionnaire to rule out any potential risk of adverse reactions to brain stimulation. All participants were right-handed according to the Edinburgh Questionnaire and had normal or corrected-to-normal vision.

**Ethics statement.**   The protocol was approved by the Ethics Committee of the Université Catholique de Louvain, Brussels, Belgium (approval number: 2018/22MAI/219) and adhered to the principles expressed in the Declaration of Helsinki. The participants were financially compensated and provided written informed consent.

## Method details

**Experimental setup.**   Experiments were conducted in a quiet and dimly lit room. Participants were seated at a table in front of a 21-inches cathode ray tube computer screen. The display was gamma-corrected, and its refresh rate was set at 100 Hz. The computer screen was positioned at a distance of 70 cm from the participants' eyes and was used to display stimuli during the decision-making task. The left and right forearms were rested upon the surface of the table with both hands on a keyboard positioned upside-down. The tip of the left and right index fingers were placed on top of the F12 and F5 keys, respectively (see Fig 1).

**Tokens task.**   The decision-making task used in the present study is a variant of the tokens task previously exploited to study decisions between reaching movements [43]; it was implemented by means of Labview 8.2 (National Instruments, Austin, Texas, United States of America). The sequence of stimuli in this task is depicted in Fig 1A. In between trials, a default screen is presented, consisting of 3 empty blue circles (4.5-cm diameter each), placed on a horizontal axis at a distance of 5.25 cm from each other. The empty circles are displayed on a white background for 2,500 ms. Each trial starts with the appearance of fifteen randomly arranged tokens (0.3-cm diameter) in the central circle. After a delay of 800 ms, a first token jumps from the center to one of the 2 lateral circles, starting the deliberation phase. The other tokens then follow, jumping one by one every 200 ms, to one of the lateral circles (i.e., 15 token jumps; $Jump_1$ to $Jump_{15}$). In this version of the task, we asked our participants to choose between left or right index finger key presses depending on which lateral circle they thought would ultimately receive the majority of the tokens (F12 or F5 key presses for left or right circle, respectively). They could choose their action and press the related key as soon as they felt sufficiently confident, as long as it was after $Jump_1$ had occurred and before $Jump_{15}$. After a choice, the tokens kept jumping every 200 ms until the central circle was empty. At this time, the circle associated with the chosen action turned either green or red depending on whether the choice was correct or incorrect, respectively, providing participants with a feedback of

their performance; the feedback also included a numerical score displayed above the central circle (see Reward, penalty, and SAT contexts section below). In the absence of any key press before $Jump_{15}$, the central circle became red and a "Time Out" message appeared on top of the screen. The feedback screen lasted for 500 ms and then disappeared at the same time as the tokens did (the circles remained on the screen), denoting the end of the trial. Each trial lasted 6,600 ms.

For each trial, we defined the "success probability" $p_i(t)$ associated with choosing each action (i.e., left or right key press) at each moment in time. If at a moment in time, the left (L) circle contains $N_L$ tokens, the right (R) one contains $N_R$ tokens, and $N_C$ tokens remain in the central (C) circle, then the probability that the left response is ultimately the correct one (i.e., the success probability of guessing left) is as follows:

$$p(L|N_L, N_R, N_C) = \frac{N_C!}{2^{N_C}} \sum_{k=0}^{min(N_C, 7-N_R)} \frac{1}{k!(N_C - k)!} \tag{1}$$

Calculating this quantity for the 15 token jumps allowed us to construct the temporal profile of success probability $p_i(t)$ for each trial. As far as the participants knew, the individual token movements and the correct choice were completely random. However, we interspersed distinct trial types within the full sequence of trials. First, in 60% of trials, the $p_i(t)$ remained between 0.5 and 0.66 up to $Jump_{10}$—i.e., the initial token jumps were balanced between the lateral circles, keeping the $p_i(t)$ close to 0.5 until late in these "ambiguous" trials. As such, in ambiguous trials, the tokens jumped alternatively to the correct and incorrect lateral circles until $Jump_{10}$, such that the number of tokens was equal in both lateral circles after each even jump (i.e., after 2, 4, 6, 8, and 10 jumps) and that a difference of one token was present after each odd jump (i.e., after 1, 3, 5, 7, and 9 jumps). Second, in 15% of trials, the $p_i(t)$ was above 0.7 after $Jump_3$ and above 0.8 after $Jump_5$—i.e., the initial jumps consistently favored the correct choice in these "obvious" trials. Then, in another 15% of trials, the $p_i(t)$ was below 0.4 after $Jump_3$—i.e., the initial jumps favored the incorrect choice and the following ones favored the correct choice in these "misleading" trials. The remaining 10% of trials were fully random (i.e., putatively involving ambiguous, obvious, and misleading trials, as well as other trials with different $p_i(t)$). Critically, the ambiguous trials were more frequent (60%) than the other trial types (30% of easy and misleading trials) because they represented our main condition of interest and their high prevalence allowed us to obtain enough probes of motor excitability during the course of deliberation in this specific setting (see TMS intensity and timings section below).

**Reward, penalty, and SAT contexts.** As mentioned above, participants received a feedback score at the end of each trial depending on whether they had chosen the correct or the incorrect circle. Correct choices led to positive scores (i.e., a reward), while incorrect choices led to negative scores (i.e., a penalty). Participants knew that the sum of these scores would turn into a monetary gain at the end of the experiment.

The reward provided for correct choices was equal to the number of tokens remaining in the central circle at the time of the key press (in € cents); hence, it gradually decreased as time elapsed in each trial (see Fig 1B). For example, a correct choice led to a reward of +10 cents when the response was provided between $Jump_5$ and $Jump_6$ (10 tokens remaining in the central circle). However, it only led to a reward of +5 cents when the response was provided between $Jump_{10}$ and $Jump_{11}$ (5 tokens remaining in the central circle). The fact that the potential reward progressively dropped produced a speed/accuracy trade-off, as participants wanted to decide fast enough to get a large reward but also slow enough to choose the correct target and avoid the penalty. This SAT has been proposed to be set by a context-dependent urgency

signal that grows over time during deliberation, as evidenced from the urgency functions obtained in such tasks [97] (see also Fig 2).

By contrast, the penalty provided for incorrect choices was constant throughout deliberation. Importantly, though, it differed in 2 block types, producing 2 SAT contexts. In the first block type, incorrect choices were severely sanctioned as the penalty there was of −14 cents, emphasizing the need for cautiousness (cautious context). Conversely, the cost of making an incorrect choice was low in the second block type as the penalty was only of −4 cents, encouraging participants to make hasty decisions in order to get high reward scores (hasty context). Hence, by manipulating the monetary cost associated with incorrect choices, we aimed at instigating distinct levels of urgency in 2 separate contexts (high and low urgency in hasty and cautious contexts, respectively), as confirmed by the analyses run on the behavioral data (please see Fig 2 and S2 Fig).

Finally, not providing a response before $Jump_{15}$ (i.e., no-response trials) also led to a penalty, which was of −4 cents both in the hasty and in the cautious contexts. Hence, in the hasty context, providing an incorrect response or not responding led to the same penalty (i.e., −4 cents), further increasing the urge to respond before the end of the trial. Conversely, in the cautious context, the potential penalty for making an incorrect choice was much higher than that obtained for an absence of response (i.e., −14 versus −4 cents, respectively), further increasing participants' cautiousness in this context.

**Time course of the sessions.** The study included 2 experimental sessions conducted at a 24-hour interval. In each session, participants realized the task in one SAT context; we thus refer to those as hasty and cautious sessions. The order of the sessions was counterbalanced across participants (see Results section for the exact counterbalancing in each analysis). Further, in order to prevent our data from being confounded by a potential difference in chronobiological states, the participants were always tested at the same time of the day [98–100].

The 2 sessions involved the same sequence of blocks. Each session started with 2 short blocks of a simple RT (SRT) task. This SRT task involves the same display as in the tokens task described above [97]. However, here, the 15 tokens remain only 50 ms in the central circle, after which they jump altogether simultaneously into one of the 2 lateral circles (always the same one in a given block). Participants were instructed to respond to this "GO signal" by pressing the appropriate key with the corresponding index finger (i.e., F12 and F5 for right and left circles, respectively). Because the circle was known in advance of the block, the task did not require any choice to be made; it was exploited to determine the participant's median SRT for left and right index finger key presses [43].

Then, participants performed training blocks to become acquainted with the tokens task. In a first training block (10 trials, only run on the first session), we ran a version of the tokens task in which the feedback was simplified; the lateral circle turned green or red, depending on whether participants had chosen the correct or the incorrect action, but no reward or penalty was provided here. Two training blocks were then realized with the full version of the task (involving rewards and penalties), one for each SAT context (20 trials each). Participants performed a last training block (20 trials), which involved the SAT context that they would be performing next during the whole session. This last block also involved TMS, which was either applied to the finger motor representation ($TMS_{Finger}$ participants) or the leg representation ($TMS_{Leg}$ participants), to prepare participants to the pulse sensation during the task. During this block, the experimenters paid particular attention to the putative presence of involuntary muscle contractions in the EMG and asked participants to relax if any contraction was detected (i.e., outside of the movement execution period), allowing them to learn to avoid preactivating their muscles during the task. No-TMS participants realized the last training block without TMS.

The actual experiment involved 8 blocks of 40 trials (regardless of choice outcome) in which participants performed the tokens task with online TMS (320 trials per session). Each block lasted about 4"30 minutes (40 trials of 6,600 ms each) and a break of 2 to 5 minutes was provided between blocks. The maximal duration of a session was 120 minutes.

**TMS over finger representations.** In $TMS_{Finger}$ participants ($n = 21$), pulses were delivered using a double-coil protocol whereby both M1 areas are stimulated at a near-simultaneous time (1-ms delay; right M1 pulse first), eliciting MEPs in finger muscles of both hands that are statistically equivalent to those obtained using classic single-coil TMS [46,101–103] (see Fig 1C and S1A and S1B Fig). Both pulses were delivered through small Fig-of-eight coils (wing internal diameter of 35 mm), which were connected to monophasic Magstim stimulators (one Magstim 200 and one Magstim Bistim[2]; Magstim, Whitland, Dyffed, United Kingdom; the side of the stimulators was counterbalanced across participants). We placed the 2 coils tangentially on the left and right side of the scalp with the handles oriented toward the back of the head and laterally at a 45˚ angle away from the midline (see Fig 1C), eliciting a current with a posteroanterior direction in the cortex.

Our objective in this group of participants was to map the spatiotemporal changes in motor excitability occurring in populations of corticospinal cells projecting to finger muscles (i.e., occurring in finger motor representations) during the index finger choices in the tokens task, in the hasty and cautious contexts. To do so, we examined MEPs in 3 different muscles, namely, the first dorsal interosseous (FDI; index finger abductor), the abductor pollicis brevis (APB; thumb abductor), and the abductor digiti minimi (ADM; pinky abductor). The FDI being prime mover in the task, its MEPs allowed us to observe excitability changes associated with a choice-relevant motor representation. As for the APB and ADM, these muscles being not required in the task, their MEPs allowed us to assess excitability changes associated with choice-irrelevant representations that lie close by the prime mover representation in the motor system (i.e., in terms of somatotopy). Importantly, MEPs in these 3 muscles were obtained by stimulating a single spot. This hotspot was found for each M1 at the beginning of every single session in each participant; it corresponded to the hotspot of the ADM [104], which usually provides the most consistent MEPs when these 3 muscles are considered together (see S1 Fig, panel A). Further, eliciting concurrent MEPs in both hands allowed us to capture excitability changes on the 2 sides of the motor system at once (in each trial), thus concerning finger representations that are both on the side of the chosen index and on the side of the unchosen finger (e.g., right and left MEPs, respectively, preceding a right index finger choice). Hence, each double-coil stimulation allowed us to obtain 6 MEPs, reflecting the excitability of 6 different finger representations playing distinct roles in the task (i.e., index, thumb, and pinky representations on the chosen and unchosen sides). The 2 M1 sites were marked on an electroencephalography cap fitted on the participant's head to provide a reference point throughout the experimental session [16,17,105].

**TMS over leg representations.** In $TMS_{Leg}$ participants ($n = 22$), TMS was applied over the leg representation of the left M1, using a batwing coil (D-B80 Magpro coil) connected to a Magpro X100 Stimulator (Magventure, Farum, Denmark). A batwing coil had to be used here because leg muscles are represented deep into the interhemispheric fissure and are difficult to target using Fig-of-eight coils, which mainly activate superficial neural layers [106]. Further, we decided to use biphasic pulses because they are known to activate deep neurons more efficiently than monophasic pulses [107,108]. The biphasic pulse was set such that its first half elicited a current with a posteroanterior direction in the cortex.

Our objective in this group of participants was to map the changes in motor excitability occurring for corticospinal cells projecting to leg muscles (i.e., in leg representations) during the index finger choices of the tokens task, in the hasty and cautious contexts. To do so, we

examined MEPs in 3 different muscles of the right leg, including the tibialis anterior (TA), as well as the lateral and medial heads of the gastrocnemius (LG and MG, respectively). These muscles being not required in the task, their MEPs allowed us to assess changes associated with choice-irrelevant representations that lie far from the prime mover representation in the motor system in terms of somatotopy. Similar as for the finger muscles, MEPs in all 3 leg muscles were obtained by stimulating a single hotspot. To do so, the coil was initially placed tangentially on the vertex of the scalp with the handle oriented toward the back of the head and parallel to the midline. Then, we turned the handle incrementally following an anticlockwise direction in order to orient the magnetic field toward the leg representation in the left M1 and to obtain maximal MEP amplitudes in the right TA muscle. Although of smaller amplitude, TMS at this location evoked consistent MEPs in the LG and MG muscles too (see S1 Fig, panel B), allowing us to broaden our observations to 2 other choice-irrelevant leg representations (see S1 Fig, panel B and D as well as S3 Fig). Here, MEPs were only obtained in right leg muscles (they were never elicited in the left leg). However, because the tokens task requires deciding between right and left index finger choices, right leg MEPs could be classified according to whether they fell on the same side as the chosen index (in right-hand trials) or on the side of the unchosen index (in left-hand trials). Hence, this design allowed us to capture excitability changes associated with leg representations of both the chosen and unchosen index sides. Similar as for the TMS$_{Finger}$ participants, the hotspot was marked on an electroencephalography cap fitted on the participant's head, providing a reference point throughout the session.

**TMS intensity and timings.** The intensity of stimulation was set in the same way in TMS$_{Finger}$ and TMS$_{Leg}$ participants. Once the hotspot was located, we first determined the individual resting motor threshold (rMT), defined as the minimal intensity required to evoke MEPs of 50 μV peak to peak on 5 out of 10 consecutive trials in the contralateral ADM or TA muscle. The ADM was used as reference in the TMS$_{Finger}$ participants because it is usually associated with a slightly higher rMT than the FDI and APB; so in this way, we obtained MEPs that are big enough in all muscles. As for the TMS$_{Leg}$ participants, setting the rMT based on the TA also allowed us to obtain reliable MEPs in the 2 other muscles.

The rMT was similar in the hasty and the cautious sessions in TMS$_{Finger}$ participants, both for the right hemisphere (45.85 ± 2.12% and 46.19 ± 1.94% of the maximum stimulator output [MSO], respectively) and for the left hemisphere (45.57 ± 1.96% and 46.23 ± 2.15% MSO, respectively). This was also the case for the rMT of left hemisphere in the TMS$_{Leg}$ participants (51.27 ± 1.73% and 51.68 ± 1.71% MSO in the hasty and the cautious sessions, respectively). In each session, TMS pulses were then applied at 120% of the rMT during the whole experiment [109].

TMS was applied both outside the blocks (i.e., at rest) and at specific timings during the blocks. MEPs elicited outside of the blocks allowed us to probe the resting-state level of motor excitability in both sessions. We recorded 20 to 25 MEPs, depending on their variability, before and after the 8 blocks of trials. Importantly, the amplitudes of these resting-state MEPs were comparable in the hasty and the cautious sessions, both in TMS$_{Finger}$ and in TMS$_{Leg}$ participants, indicating that the stimulation protocol guaranteed reproducible measurements across experimental sessions (see S1 Fig, panel B and D for details).

When applied during the blocks, TMS could occur at 1 of 4 different timings (see Fig 1C), randomized within each session. First, it could occur when the circles were empty (i.e., 500 ms before the appearance of the tokens in the central circle), allowing us to measure the baseline level of motor excitability while participants were at rest but engaged in the task. Moreover, TMS could occur at 1 of 3 different token jumps: Jump$_1$, Jump$_4$, or Jump$_7$ (i.e., corresponding to 0, 600, and 1,200 ms from deliberation onset). The MEPs recorded at these timings served to probe the changes in motor excitability during the deliberation process.

**Table 1. Number of trials for each trial type and TMS timing.**

| | Nonstimulated | Stimulated | | Total |
|---|---|---|---|---|
| | | Baseline | Deliberation | |
| Ambiguous | 17 | 0 | **170**[*] | 187 |
| Obvious | 3 | 0 | 46 | 49 |
| Misleading | 6 | 0 | 48 | 54 |
| Random | 3 | **27** | 0 | 30 |
| **Total** | 29 | 27 | 264 | 320 |

The 2 numbers highlighted in bold represent the trials exploited in our MEP analysis. Stimulation at baseline occurred in 27 of the 30 random trials. Further, 170 ambiguous trials were stimulated during deliberation, allowing us to probe changes in motor excitability during the decision process.

[*]Among the 170 MEPs that were elicited in ambiguous trials, 58 were obtained at $Jump_1$, 56 at $Jump_4$, and 56 at $Jump_7$.

MEP, motor-evoked potential; TMS, transcranial magnetic stimulation.

MEPs were elicited in about 90% of the total number of trials in both contexts ($n = 291/320$). Hence, about 10% of trials did not involve TMS (about 4 trials per block), preventing participants from anticipating the stimulation. Further, the percentage of stimulated trials was the same across trial types (i.e., 90% of ambiguous, obvious, misleading, and random trials), such that participants could not associate a particular trial type with TMS. However, the MEPs elicited in obvious and misleading trials ($n = 94$ in total, including all TMS timings) were not exploited in our analyses, as those usually involved responses before $Jump_7$ or even before $Jump_4$ [84]. In contrast, ambiguous trials, which were our focus of interest, typically led to participants responding after $Jump_7$. To ensure such long RTs in ambiguous trials, the distribution of tokens was kept relatively balanced between the 2 lateral circles until $Jump_{10}$, yielding no real fluctuation in sensory evidence before that and thus at the times TMS was applied (i.e., $Jump_1$, $Jump_4$, and $Jump_7$). Hence, changes in motor activity cannot be accounted for by variations in sensory evidence [110–112]. In total, for each session (i.e., in each context), 170 MEPs were elicited in ambiguous trials, among which 58 were obtained at $Jump_1$, 56 at $Jump_4$, and 56 at $Jump_7$. Baseline MEPs were elicited in the random trials ($n = 27$). This large number of trials allowed us to account for potential within-participant variability in MEP amplitudes. Table 1 below synthesizes the number of nonstimulated trials as well as trials stimulated at baseline and during deliberation.

**EMG data collection.** Surface EMG electrodes (Medicotest, USA) were placed on the investigated muscles (i.e., right and left FDI, APB and ADM in $TMS_{Finger}$ participants, and right TA, LG and MG in $TMS_{Leg}$ participants), allowing us to record the MEPs elicited in these muscles. The EMG signals recorded in $TMS_{Finger}$ participants were also exploited to quantify movement vigor (see below). The ground electrode was placed over the right ulnar styloid process in the $TMS_{Finger}$ participants and over the right patella in the $TMS_{Leg}$ participants. The signals were recorded for 4,000 ms on each trial, starting 500 ms before the first TMS timing (i.e., before baseline) and ending 1,000 ms after the last TMS timing (i.e., after $Jump_7$). The EMG signals were amplified, band-pass filtered (10 to 500 Hz) and notch filtered (50 Hz) online (NeuroLog, Digitimer, UK), and digitized at 2,000 Hz for offline analysis. The experimenters visually screened the signals throughout the acquisitions and asked participants to relax if any contraction was apparent.

## Quantification and statistical analysis

Behavioral data were collected by means of LabView 8.2 (National Instruments), stored in a database (Microsoft SQL Server 2005, Redmond, Washington, USA), and analyzed with

ccustom Matlab (MathWorks, Natick, Massachusetts, USA) and R scripts (R Core Team, 2020). EMG data were collected using Signal 6.04 (Cambridge Electronic Design, Cambridge, UK) and analyzed with custom Signal and R scripts. Statistical analyses were performed using custom R scripts and Statistica 7.0 (StatSoft, Oklahoma, USA).

**Decision behavior quantification.** For each participant and each SAT context, we computed the median DT (all trial types pooled together) and decision accuracy (i.e., percentage of correct choices over total number of choices made). To estimate the DT in each trial, we first calculated the RT during the tokens task by computing the difference between the time at which the participant pressed the key and the time of $\text{Jump}_1$. We then subtracted from the single-trial RTs the median SRT for each participant (i.e., difference between key press and the tokens' jump in the SRT task). This procedure allowed us to remove from the individual RT obtained in the tokens task, the sum of the delays attributable to sensory processing of the stimulus display as well as to response initiation and muscle contraction, providing us the DT [43].

The tokens task also allowed us to estimate the amount of evidence based on which participants made their action choices in each SAT context. To do so, we first computed a first-order approximation of the real probability function after each jump (see Eq 1), called the sum of log-likelihood ratios (SumLogLR) [43,94]:

$$SumLogLR(n) = \sum_{k=1}^{n} log \frac{p(e_k|C)}{p(e_k|U)} \qquad (2)$$

In this equation, $p(e_k|C)$ is the likelihood of a token event $e_k$ (a token favoring either the chosen or the unchosen action) during trials in which the chosen action $C$ is correct, and $p(e_k|U)$ is the likelihood of $e_k$ during trials in which the unchosen action $U$ is correct. The Sum-LogLR is proportional to the difference between the number of tokens that favored each of the 2 possible choices (i.e., that moved toward each lateral circle) at any given time. Hence, the lower the amount of sensory evidence in favor of the chosen action, the lower the SumLogLR. To characterize the decision policy of the participants in each SAT context, we determined the level of sensory evidence at the time of commitment as a function of the participant's DT. To do so, we grouped the trials into 10 consecutive percentile bins of DT ($DT_{Bin1-10}$) and then calculated the average SumLogLR corresponding to each DT bin in each participant. Note that for this analysis, we consider a simplified scenario where the commitment time is estimated as the end of the DT, which neglects the duration required for sensory processing.

We exploited the obtained SumLogLR at DT values to estimate urgency functions. As such, models of decision-making that incorporate an urgency signal, posit that choices result from the combination of signals that reflect the available sensory evidence and the level of urgency that grows over time (e.g., [60,61]). For example, in a minimal implementation of the urgency-gating model [43,94], evidence is multiplied by a linearly increasing urgency signal and then compared to a fixed decision threshold. The result can be expressed as follows:

$$y_i = (N_i - N_{j \neq i}) \cdot [at + b]^+ < T, \qquad (3)$$

where $y_i$ is the "neural activity" for action choices to lateral circle $i$, $N_i$ is the number of tokens in lateral circle $i$, $t$ is the number of seconds elapsed since the start of the trial, $a$ and $b$ are the slope and y-intercept of the urgency signal, and $[]+$ denotes half-wave rectification (which sets all negative values to zero). When $y_i$ for any action crosses the threshold $T$, that action is chosen.

A direct implication of such urgency-based models is that decisions made with low levels of sensory evidence should be associated with high levels of urgency and vice versa. That is, one

core assumption is that a high urgency should push one to commit to a choice even if evidence for that choice is weak. Hence, considering a model in which evidence is multiplied by an urgency signal, we estimated urgency values based on the SumLogLR at DT obtained in each participant, at each bin, and in each SAT context, as follows:

$$U_{(s,t,c)} = \frac{T}{SLR_{(s,t,c)}} \tag{4}$$

In the above, $s$ is the participant number, $t$ is the DT bin, $c$ is the SAT context, $SLR$ is the SumLogLR at DT, $T$ is a constant representing a fixed threshold (which we fixed to 1), and $U$ is the estimated urgency value. We then fitted a linear regression model over the obtained urgency values and extracted the intercept and the slope of the functions for each participant and both contexts.

**Movement vigor quantification.** We also examined the vigor with which the participants pressed the response key in the hasty and cautious contexts. To do so, we exploited the EMG signals recorded in the finger muscles in $TMS_{Finger}$ participants (i.e., in left and right index, thumb, and pinky muscles) and considered the magnitude of EMG burst preceding the key press as a proxy of movement vigor [2,68].

First, the signals were segmented into epochs extending from −300 to 0 ms with respect to the key press (i.e., 600 data points). Trials in which a TMS pulse occurred between −400 and 0 ms were discarded from the analysis, preventing contamination of the segmented signals from TMS artifacts and MEPs. For each epoch, we then removed any putative signal offset by subtracting the average signal amplitude in the first 50 ms from every data point of the epoch. The signals were subsequently rectified by taking the absolute value of each data point.

In a following processing step, we classified the epochs according to the individual's RT in the trial, allowing us to test for any impact of elapsed time on movement vigor (in addition to the impact of context). Epochs were categorized depending on whether they were associated with a short or a long RT using a median-split approach ($RT_{Short}$ or $RT_{Long}$, respectively). Further, given the expected between-context difference in RTs and its potential effect on movement vigor [21,94], we adopted a RT-matching procedure to homogenize $RT_{Short}$ and $RT_{Long}$ distributions across contexts [1]. The procedure consisted in discretizing each participant's $RT_{Short}$ and $RT_{Long}$ distributions into bins of 200 ms width and, for each bin, randomly selecting a matched number of trials from the context condition that had the greatest trial count in that bin. One participant had to be discarded from the analysis at this step because the overlap between his/her RT distributions across contexts was too small, leaving less than 6 trials for some conditions after the matching procedure. The remaining 20 participants presented an average of 62 ± 2 trials per $RT_{Length}$ and context (range: [50 to 73 trials]). Following this step, the trials included in the analysis involved homogenous $RT_{Short}$ and $RT_{Long}$ across the hasty and cautious contexts, as depicted in S5 Fig ($RT_{Short}$: 1,417 ± 46 ms and 1,422 ± 46 ms, respectively; $RT_{Long}$: 2030 ± 20 ms and 2,034 ± 19 ms, respectively).

We then computed the median value of each data point across the epochs for each condition of interest, providing us with 24 signals per participant: That is, one signal was obtained for each muscle (index, thumb, and pinky muscles), each hand (chosen and unchosen), each $RT_{Length}$ ($RT_{Short}$, $RT_{Long}$) and each SAT context ($Context_{Hasty}$, $Context_{Cautious}$). These signals were baseline corrected (i.e., baseline subtraction; reference window: −300 to −200 ms) and low-pass filtered (butterworth filter; order: 1, cutoff frequency: 5 Hz). Three variables were finally extracted to quantify movement vigor in each condition in the chosen hand: the maximal peak amplitude and the time-to-peak amplitude. The latest variable was estimated by computing the difference between the maximal peak timing and the onset of voluntary contraction

(estimated using a threshold of 3 standard deviation [SD] above the average signal amplitude in a window extending from −300 to −200 ms).

**Motor excitability quantification.** Motor excitability was quantified based on the absolute peak-to-peak amplitude of MEPs (in μV) in each target muscle of the $TMS_{Finger}$ and $TMS_{Leg}$ participants. As mentioned above, MEPs elicited at $Jump_1$, $Jump_4$ and $Jump_7$ were only considered in ambiguous trials. Moreover, we only included trials in which the RT was comprised between 1,350 and 2,800 ms (i.e., at least 150 ms after $Jump_7$ and up to $Jump_{15}$; see S4 Fig). Hence, even in trials with TMS at the latest time point ($Jump_7$), the selected trials involved MEPs that fell relatively far from movement onset (at least 150 ms before the key press), allowing us to capture motor excitability changes that are specific to deliberation and not movement execution. Both correct and incorrect trials were included in the MEP analysis. As such, the proportion of incorrect trials was too low to consider them separately in our MEP analyses. No-response trials were excluded from the analysis.

In order to prevent contamination of the measurements from background muscular activity, participants were reminded to relax during the whole experiment based on the EMG signals, which were continuously screened by the experimenters. Furthermore, as mentioned above, participants initially performed a training block during which they were provided feedback on whether they were adequately relaxed or not. Moreover, trials in which the root mean square of the EMG signal exceeded 3 SD above the mean before stimulation (i.e., −250 to −50 ms from the pulse) were discarded from the analyses (rejection rate: 8.48 ± 0.43% in $TMS_{Finger}$ participants and 0.99 ± 0.05% in $TMS_{Leg}$ participants). Finally, to attenuate the effect of MEP variability on our measures, MEPs with an amplitude exceeding 3 SD around the mean were excluded too (rejection rate: 3.70 ± 0.42% in $TMS_{Finger}$ participants and 2.37 ± 0.24% in $TMS_{Leg}$ participants).

Following this cleaning procedure in the $TMS_{Finger}$ participants, we had 39 ± 0.3 and 38.8 ± 0.2 trials left with TMS falling outside of the blocks (i.e., at rest) in the hasty and cautious sessions, respectively. For the analysis of motor excitability in these resting-state trials, we first computed separate medians of MEP amplitude for each hand, and this, for each finger representation, each SAT context and each participant. We then further pooled the MEP data from the left and right hands to obtain a single resting-state motor excitability value for each finger representation, context, and participant. In the $TMS_{Leg}$ participants, we were left with 46.2 ± 0.4 and 45.4 ± 0.3 resting-state trials in the 2 corresponding sessions. Here, MEPs were only elicited in the right leg, and separate medians were computed for each motor representation (i.e., of the TA, MG, and LG muscles), each context and each participant.

Trials in which MEPs occurred at $Jump_1$, $Jump_4$ and $Jump_7$ were further processed using a RT-matching procedure, allowing us to homogenize RT distributions across contexts for each TMS timing separately (see S2 Fig). Following this step, 2 $TMS_{Finger}$ and 6 $TMS_{Leg}$ participants had to be discarded from the analysis because their datasets fell to less than 8 trials on average across TMS timings (the behavioral data and the baseline and resting-state MEP data of these participants were conserved in the respective analyses). The datasets of the remaining 19 $TMS_{Finger}$ and 16 $TMS_{Leg}$ participants comprised an average of 28 ± 2 and 15 ± 1 trials, respectively, across TMS timings and SAT contexts (range: [10 to 39 trials] and [8 to 27 trials]). The included trials involved comparable RTs in hasty and cautious contexts, both in $TMS_{Finger}$ participants (2,230 ± 39 ms and 2,230 ± 38 ms, respectively) and in $TMS_{Leg}$ participants (2,243 ± 28 ms and 2,249 ± 26 ms, respectively).

Preliminary analyses showed that, if performed multiple times, the trial selection of the matching procedure could produce subtle variations in MEP amplitudes when trials were then pooled across conditions (e.g., across TMS timings, contexts, etc.), depending on which trials were eventually included in the analysis. Hence, to avoid any effect of the trial selection on the

results, the procedure was repeated 100 times, the median MEP amplitude was first calculated for each condition and for every iteration, and we then calculated the median MEP amplitude across iterations (see [1] for a similar procedure). Following this step in $TMS_{Finger}$ participants, one MEP amplitude was obtained for 72 conditions, namely for each TMS timing ($Jump_1$, $Jump_4$, and $Jump_7$), each context (hasty and cautious), and each of the 6 motor representations (left and right FDI, APB, and ADM), when these representations were classified as part of the chosen or the unchosen side of the motor system. In $TMS_{Leg}$ participants, one MEP amplitude was obtained for each TMS timing ($Jump_1$, $Jump_4$, and $Jump_7$), each context (hasty and cautious), each of the 3 motor representations (right TA, LG, and MG), and each side (chosen and unchosen). Besides, baseline MEP amplitudes were not subjected to the RT-matching procedure and were directly pooled together for each context and each representation, independently of the side that ended up being chosen.

Once the median MEP amplitudes were obtained (in μV), we normalized them (in %). That is, MEPs obtained at baseline were expressed in percentage of resting-state amplitudes [63,113], providing us with a normalized measure of baseline excitability for each motor representation and each context. Further, amplitudes obtained at $Jump_1$, $Jump_4$, and $Jump_7$ were expressed in percentage of baseline amplitudes [75,114], providing a normalized measure of excitability for each motor representation on the side of the chosen and unchosen index fingers, in each context. Notably, in $TMS_{Finger}$ participants, we first normalized separately MEPs associated with left and right finger representations and then pooled the obtained values together according to whether they fell on the side of the chosen or the unchosen index finger.

Ultimately, we computed spatiotemporal maps to provide an integrative view of motor excitability changes occurring during the course of deliberation in each context. To this aim, we considered the MEPs obtained for the index (FDI), thumb (APB), pinky (ADM), and leg (TA, LG, and MG pooled together) representations on the side of both the chosen and unchosen index fingers (i.e., 8 representations). For each representation in each context, we averaged excitability across participants and then performed a linear interpolation to estimate excitability changes between each timing (100 data points between each timing), providing us with a temporally continuous trace. For each context, the 8 traces were then spatially arranged according to M1 somatotopy: that is, traces of the thumb, index, pinky, and leg representations on the chosen side (i.e., lateromedial arrangement) were followed by traces of the unchosen leg, pinky, index, and thumb representations (i.e., mediolateral arrangement). Here again, a linear interpolation was performed to estimate excitability changes between each representation (100 data points), providing us with a spatially continuous trace at each time point. Two spatiotemporal maps were thus obtained (one for each context) and a between-context difference map was finally computed (i.e., hasty minus cautious context).

**Single-trial correlation of motor excitability between the chosen index and other finger representations.** In the $TMS_{Finger}$ participants, the use of double-coil TMS allowed us to obtain MEPs from 6 finger muscles at once in each trial. Hence, besides considering the amplitude of MEPs within each of these muscles separately, we could also assess the degree to which MEPs in these different muscles varied in concert from one trial to another, providing us with a measure of their relationship in terms of changes in motor excitability. Here, we focused on the link between the chosen index finger and each of the 5 other finger representations. To do so, we exploited an approach inspired by seed-based correlation analyses (SCAs), which are usually applied on neuroimaging data to quantify correlations between activity changes in a specific region of interest (i.e., the seed) and other brain regions (e.g., [115–117]). For the purpose of this study, we defined the representation of the chosen index finger as our seed and quantified the relationship between this key representation and each of the 5 other finger representations (i.e., thumb and pinky on the same (chosen) side, as well as index, thumb, and

pinky on the unchosen side). The rationale here was that a high positive correlation between the index and the other finger muscles would indicate the operation of influences exerting a broad, common impact on their motor representations [52,53], shaping MEPs in block. In contrast, a low or even a negative correlation would indicate the presence of influences affecting the chosen index representation in a more selective and differentiated way. We were interested in comparing the strength of the bond linking the chosen index to the other fingers between both contexts.

To this aim, we exploited the single-trial MEPs obtained at $Jump_7$ following the 100 iterations of the RT-matching procedure described above. These single-trial MEPs were normalized as a percentage of the average baseline amplitude for each finger representation and each context in each participant. Importantly, we considered the trials of all participants ($N_{Participants}$ = 19), providing us with a large pool of data points ($N_{Trials}$ = 528). Given the RT-matching procedure, the number of trials for a given participant was equal in each context, such that each one had the same weight in each correlation. To normalize distribution of the single-trial data, we applied a square root transformation on each data point (the findings presented in Fig 6 still hold without this transformation).

A rmCorr was exploited using the rmcorr package in R [65]. RmCorr is a statistical technique for determining the common within-individual association for paired measures assessed on 2 or more occasions for multiple individuals. As such, simple regression/correlation is often applied to nonindependent observations or aggregated data; this may produce biased, specious results due to violation of independence and/or differing patterns between participants versus within-participants [118]. Unlike simple regression/correlation, rmCorr does not violate the assumption of independence of observations [65]. The approach accounts for nonindependence among observations using ANCOVA to statistically adjust for interindividual variability. By removing measured variance between participants, rmCorr provides the best linear fit for each participant using parallel regression lines (the same slope) with varying intercepts. Hence, rmCorr tends to have much greater statistical power because neither averaging nor aggregation is necessary for an intraindividual research question. rmCorr estimates the common regression slope, the association shared among individuals. Conceptually, rmCorr is close to a null multilevel model (i.e., varying intercept and a common slope for each individual), but the techniques differ on how they treat/pool variance. RmCorr assesses the common intraindividual variance in data, whereas multilevel modeling analyzes simultaneously different sources of variance using fixed and random effects.

The rmCorr procedure was repeated 100 times (i.e., corresponding to the 100 pools of data points obtained following the RT-matching procedure), providing us with 100 R-values and 100 permutation-based $p$-values. We finally calculated the median of these R- and $p$-values across iterations as estimate values of the correlations. Given that 10 correlations were performed (i.e., 5 representation pairs in both contexts), the significance threshold was set at 0.005 after Bonferroni correction. Finally, we looked for any difference in those R-values between contexts using a comparison of 95% CIs (see Statistical analysis section below).

**Statistical analysis.** No-TMS, $TMS_{Finger}$, and $TMS_{Leg}$ participants all exhibited strongly similar decision behavior (presented in detail in S2 Fig). Hence, the behavioral data of the 50 participants were considered altogether in a single statistical analysis (Fig 2). First, a permutation-based Pearson correlation was realized to test any significant relationship between DTs and decision accuracy in each context ($N_{Permutations}$ = 1,000). The DT, decision accuracy, urgency slope, and intercept data were then compared across contexts using 2-tailed Student $t$ tests for paired-samples. For each context, the slope of the urgency function was further compared against 0 using a 2-tailed $t$ test. Effect sizes were estimated for each $t$ test by calculating Cohen's d values. In accordance with conventional interpretation of Cohen's d, a value of 0.2

is interpreted as indicating a small effect size, a value of 0.4 a medium effect size, and a value of 0.8 or more as a large effect size [119].

Most of the ensuing statistical comparisons involved repeated measures analyses of variance (rmANOVAs). When performing rmANOVAs, Maunchley tests were exploited systematically to check for data sphericity and GG corrections were used to correct for any deviation from sphericity. Post hoc comparisons were conducted using the Tukey HSD procedure. Effect sizes were estimated for each main effect and interaction by calculating partial eta squared ($\eta^2$). In accordance with conventional interpretation partial $\eta^2$, a value of 0.01 is interpreted as indicating a small effect size, a value of 0.06 a medium effect size, and a value of 0.14 or more as a large effect size [120].

The effect of elapsed time and context on movement vigor was tested using 3-way rmANOVAs on the maximal peak amplitude and the time-to-peak amplitude data with MUSCLE (index, thumb, and pinky muscles), $RT_{LENGTH}$ ($RT_{Short}$ and $RT_{Long}$), and CONTEXT (hasty and cautious) as within-participant factors. We performed this analysis post hoc, once the study completed, in order to check for any putative role of vigor on the motor excitability changes observed in the dataset.

Normalized excitability data obtained at baseline were analyzed using 2-way rmANOVAs with REPRESENTATION (index, thumb, and pinky in $TMS_{Finger}$ participants and tibialis, lateral and medial gastrocniemius in $TMS_{Leg}$ participants) and CONTEXT (hasty and cautious) as within-participant factors. In addition, excitability data measured during deliberation on the side of the chosen and unchosen index fingers were analyzed using 2 separate 3-way rmANOVAs with TIMING ($Jump_1$, $Jump_4$, and $Jump_7$), REPRESENTATION, and CONTEXT as within-participant factors.

When a rmANOVA pointed to a lack of significant effect, a BF analysis was performed to quantify statistically the level of evidence for a lack of effect. These analyses were also decided post hoc by exploiting the BayesFactor package in R, using the default settings. BFs provided us with a ratio of the likelihood probability of the null hypothesis (i.e., H0: the probability that data do not exhibit an effect of factor tested) over the alternative hypothesis (i.e., H1: the probability that data exhibit the effect [64]). A BF value of 1 would reflect an equal probability that H0 and H1 are correct, whereas a BF value higher than 1 would reflect a higher probability that H0 is correct. In accordance with conventional interpretation of BF values [121], a BF ranging between 1 and 3 is interpreted as indicating anecdotal evidence in favor of H0, a value between 3 and 10 as indicating substantial evidence for H0, a value between 10 and 100 a strong evidence for H0, and a value above 100 a decisive evidence for H0.

Finally, we tested the effect of context on the single-trial correlations. As such, the rmCorr $p$-values allowed us to identify changes in the significance of the correlations between contexts. However, in order to quantify such changes more directly, we compared the strength of the correlation between contexts using a direct comparison of the 95% CI calculated using the rmCorr approach. A difference in R-values between contexts was considered as significant when the 95% CIs for the compared R-values did not overlap.

## Supporting information

**S1 Fig. (related to Fig 1C). Resting-state MEPs. (A)** Resting-state recordings in a representative $TMS_{Finger}$ participant. As described in the Materials and methods section, participants performed the 2 block types (i.e., hasty and cautious context blocks) on separate experimental sessions. The blue and yellow traces represent raw MEP recordings, as obtained at rest in the hasty and cautious context sessions, respectively. In each session, the double-coil stimulation over the left and right finger representations allowed us to elicit MEPs in the index, the thumb

and the pinky muscles of both hands at once. **(B)** Same as A for a $TMS_{Leg}$ participant. MEP amplitudes were smaller in the leg than in the finger representations, potentially due to the higher distance between the coil and the leg area, located in the interhemispheric fissure. Still, in each session, the stimulation over the left leg representations allowed us to elicit MEPs of reliable amplitudes in the right TA, as well as in the right lateral and medial heads of the gastrocnemius muscle at once. **(C, D)** Group-averaged resting-state excitability and statistical analysis. NS annotations indicate that the [rm]ANOVAs performed on resting-state MEPs did not show any significant difference between the hasty and cautious sessions, neither in $TMS_{Finger}$ participants (Effect of SESSION: $F_{1,20} = 0.48$, $p = 0.497$, partial $\eta^2 = 0.023$; SESSION*REPRESENTATION interaction: $F_{2,40} = 1.08$, $p = 0.348$, partial $\eta^2 = 0.051$), nor in $TMS_{Leg}$ participants (Effect of SESSION: $F_{1,21} = 0.19$, $p = 0.663$, partial $\eta^2 = 0.009$; SESSION*REPRESENTATION interaction: $F_{2,42} = 2.07$, $p = 0.138$, partial $\eta^2 = 0.089$). Further, a BF analysis provided substantial evidence for a lack of effect of the factor SESSION on resting-state MEPs (BFs = 5.58 and 5.30, in $TMS_{Finger}$ and $TMS_{Leg}$ participants, respectively). The hash signs above the bars indicate that MEP amplitudes were significantly higher than 0 in all muscles (all t-values > 5.5, all $p$-values < 0.0001 after Bonferroni correction). Error bars represent 1 SEM. All individual and group-averaged numerical data exploited for S1 Fig are freely available at this link: https://osf.io/tbw7h. Altogether, these data show that the 2 TMS protocols allowed us to record MEPs that were both reproducible across sessions and of reliable amplitudes in all of the investigated muscles. BF, Bayes factor; MEP, motor-evoked potential; rmANOVA, repeated measures analyses of variance; TA, tibialis anterior; TMS, transcranial magnetic stimulation.
(DOCX)

**S2 Fig. (related to Fig 2). The context-dependent shift in decision behavior was comparable in the 3 TMS subgroups (i.e., $TMS_{Finger}$ [top panel] and $TMS_{Leg}$ [middle panel] and No-TMS participants [bottom panel]). (A)** DTs. We performed a rmANOVA while considering TMS-SUBGROUP as a categorical predictor. We did not find any significant effect of TMS-SUBGROUP ($F_{2,47} = 1.39$, $p = 0.258$, partial $\eta^2 = 0.056$) nor of its interaction with the factor CONTEXT on DTs ($F_{2,47} = 1.03$, $p = 0.362$, partial $\eta^2 = 0.042$). Further, a BF analysis provided substantial evidence for a lack of effect of the TMS-SUBGROUP on DTs (BF = 4.06). **(B)** Same as A. for decision accuracy. There was no significant effect of TMS-SUBGROUP ($F_{2,47} = 1.05$, $p = 0.357$, partial $\eta^2 = 0.043$) nor of its interaction with the factor CONTEXT on accuracy ($F_{2,47} = 0.38$, $p = 0.687$, partial $\eta^2 = 0.016$). The BF was of 4.07 for the effect of the TMS-SUBGROUP, revealing substantial evidence for a lack of effect of this factor on accuracy. **(C)** Urgency functions. There was also no significant effect of TMS-SUBGROUP ($F_{2,47} = 0.89$, $p = 0.415$, partial $\eta^2 = 0.037$) nor of its interaction with the factor CONTEXT on the slope of the urgency functions ($F_{2,47} = 0.81$, $p = 0.452$, partial $\eta^2 = 0.033$). Similarly, there was no significant effect of TMS-SUBGROUP ($F_{2,47} = 0.19$, $p = 0.820$, partial $\eta^2 = 0.008$) nor of its interaction with the factor CONTEXT on the intercept of the functions ($F_{2,47} = 0.44$, $p = 0.643$, partial $\eta^2 = 0.018$). Here again, BFs showed substantial evidence for a lack of effect of the TMS-SUBGROUP on the slope and the intercept of the functions (BFs = 4.22 and 8.35, respectively). Error bars represent 1 SEM. All individual and group-averaged numerical data exploited for S2 Fig are freely available at this link: https://osf.io/tbw7h. Overall, these results highlight that the 3 subgroups presented very similar effects of context on all of these behavioral variables, indicating that the application of TMS over the finger and leg representations did not perturb SAT regulation in our task. DT, decision time; rmANOVA, repeated measures analyses of variance; SAT, speed–accuracy trade-off; TMS, transcranial magnetic stimulation.
(DOCX)

**S3 Fig. (related to Fig 3). Trial selection and RT-matching procedure for MEP analysis.**
Given the between-context difference in decision speed (see Fig 2 and S1 Fig) and its potential effect on MEP amplitudes, we adopted a RT-matching procedure to homogenize RT distributions across contexts (see [1] for a similar procedure). The procedure consisted in discretizing each participant's RT distributions into bins of 200 ms width and, for each bin, randomly selecting a matched number of trials. To do so, for each bin, we kept all the trials of the context condition that had the lowest trial count and selected a matched number from the context condition that had the greatest trial count in that bin. As a result, the MEPs included in the analysis were those for which the RT distributions for the hasty and cautious overlapped (gray area on single-participant distributions). In a few participants (for whom the distribution overlap was very small), this procedure led to the exclusion of many trials. Hence, we had to exclude 2 and 6 participants out of the 21 $TMS_{Finger}$ and 22 $TMS_{Leg}$ participants as they presented too few trials after this procedure for each timing and context (i.e., <8 trials on average). On the remaining 19 $TMS_{Finger}$ and 16 $TMS_{Leg}$ participants, the included trials involved comparable RTs in the hasty and cautious contexts, both in $TMS_{Finger}$ participants (2,230 ± 39 ms and 2,230 ± 38 ms, respectively) and in $TMS_{Leg}$ participants (2,266 ± 27 ms and 2,278 ± 26 ms, respectively; see bar graphs). Error bars represent 1 SEM. All individual and group-averaged numerical data exploited for S3 Fig are freely available at this link: https://osf.io/tbw7h. Hence, this procedure guaranteed that any effect of context on MEP amplitudes could not result from a between-context difference in RT in the included trials. MEP, motor-evoked potential; RT, reaction time; TMS, transcranial magnetic stimulation.
(DOCX)

**S4 Fig. (related to Fig 4). The context-dependent shift in decision behavior was present in the participants included in the MEP analysis.** As mentioned in the Results and in the Materials and methods sections, we had to exclude 3 out of the 21 $TMS_{Finger}$ participants and 6 out of the 22 $TMS_{Leg}$ participants following the RT-matching procedure. Participants excluded following this procedure were more likely to present a too small overlap between their RT distributions and thus to exhibit strong SAT shifts. To ensure that the participants included in the MEP analysis presented a SAT shift, we performed a statistical analysis on their behavioral data. A between-context comparison revealed that DTs and accuracy were significantly lower in the hasty context ($TMS_{Finger}$ and $TMS_{Leg}$ participants pooled together: $t_{34} = -7.69$, $p < 0.0001$, Cohen's d = 1.304 and $t_{34} = -9.25$, $p < 0.0001$, Cohen's d = 1.565, respectively; panel A and B). Besides, the urgency intercept was significantly higher in the hasty relative to the cautious context ($t_{49} = 5.37$, $p < 0.0001$, Cohen's d = 0.909; panel C). All individual and group-averaged numerical data exploited for S4 Fig are freely available at this link: https://osf.io/tbw7h. DT, decision time; MEP, motor-evoked potential; RT, reaction time; SAT, speed–accuracy trade-off; TMS, transcranial magnetic stimulation.
(DOCX)

**S5 Fig. (related to Fig 4). The broad amplification affected the 3 leg representations of the chosen side in a reproducible way. (A)** Effect of CONTEXT on motor excitability on the chosen side. Excitability changes were more variable in the leg than in the finger representations (i.e., compared to Fig 4), potentially due to the smaller MEP amplitudes obtained for the leg representation (see S1 Fig). Despite this variability, the effect of context was comparable in the 3 investigated leg representations, with higher excitability values in the hasty than in the cautious context. As such, there was no significant CONTEXT*REPRESENTATION interaction ($F_{2,30} = 0.53$, $p = 0.595$, partial $\eta^2 = 0.034$), nor any CONTEXT*TIMING*REPRESENTATION interaction ($F_{4,60} = 1.53$, $p = 0.202$, partial $\eta^2 = 0.093$); BF for the latter analysis was of 14.03, providing strong evidence for a lack of effect on this interaction. *: significant effect of context

at $p < 0.05$. **(B)** Same as A. for the unchosen side. Here again, the 3 leg representations exhibited similar patterns of excitability changes, with no evident impact of context and an overall rise as time elapsed. Indeed, there was no significant CONTEXT*REPRESENTATION interaction ($F_{2,30} = 0.58$, $p = 0.561$, partial $\eta^2 = 0.037$), nor any CONTEXT*TIMING*REPRESENTATION interaction ($F_{4,60} = 0.47$, $p = 0.756$, partial $\eta^2 = 0.030$); BF for the latter analysis was of 17.65, providing strong evidence for a lack of effect on this interaction. Error bars represent 1 SEM. All individual and group-averaged numerical data exploited for S5 Fig are freely available at this link: https://osf.io/tbw7h. BF, Bayes factor; MEP, motor-evoked potential.
(DOCX)

**S6 Fig. (related to Fig 4). The effects of context were still present when exploiting the full, RT-unmatched dataset.** The RT-matching procedure described in S3 Fig ensured similar RTs between the 2 contexts, but it raises a potential confound by emphasizing the slowest trials from the hasty context and the fastest trials from the cautious context. However, concerns about that confound are reduced by the observation that the same analyses performed on the full set of trials, without RT matching, produced the same results. All individual and group-averaged numerical data exploited for S6 Fig are freely available at this link: https://osf.io/tbw7h. RT, reaction time.
(DOCX)

**S7 Fig. (related to Fig 6). Single-trial correlations.** Example of 4 correlations obtained from the single-trial analysis. We pooled the trials of all participants together (normalized to baseline; $N_{Participants} = 16$), providing us with a large pool of data points ($N_{Trials} = 528$) and applied a rmCorr analysis. RmCorr accounts for nonindependence among observations using ANCOVA to statistically adjust for interindividual variability. By removing measured variance between participants, rmCorr provides the best linear fit for each participant using parallel regression lines (the same slope) with varying intercepts, as can be seen in each cloud of points. As indicated in the Materials and methods section, normalized single-trial data were squared root-transformed to enhance the normality of the distributions (although similar findings were obtained on nontransformed data). As evident on this figure, excitability changes in the chosen index representation positively covaried with changes in other finger representations (here, the thumb and pinky representations of the chosen side). Further, while the strength of the correlation between the index and the thumb representation was comparable in the hasty and cautious contexts (R = 0.63 and 0.56, 95% CIs = [0.58 0.68] and [0.49 0.61], respectively), the correlation between the index and the pinky representation was significantly weaker in the former than the latter context (R = 0.08 and R = 0.34, 95% CIs = [−0.007 016] and [0.27 0.42], respectively). A similar decorrelation was observed between the chosen index and the index and thumb representations of the unchosen side (see Fig 6 in the main text). All individual and group-averaged numerical data exploited for S7 Fig are freely available at this link: https://osf.io/tbw7h. ANCOVA, analysis of covariance; rmCorr, repeated measures correlation.
(DOCX)

**S8 Fig. RT median split and RT-matching procedures on movement vigor data. (A)** Distributions of included trials. In order to investigate the effect of elapsed time on this EMG peak amplitude in each context and in every participant, we split the trials into 2 subsets according to whether they were associated with short or long RTs, using a median-split procedure ($RT_{Short}$ and $RT_{Long}$ trials, left and right panels, respectively). Further, to prevent EMG peak amplitude from being affected by the difference in decision speed between each context, we homogenized the RT distributions across contexts through a RT-matching procedure (same as

described in S3 Fig on MEP data), both for $RT_{Short}$ and $RT_{Long}$ trials. **(B)** Group-averaged EMG peak amplitude. Following the RT-matching procedure, we had to exclude 1 out of the 21 $TMS_{Finger}$ participants, as he/she ended up with no trial in a specific condition. As evident on the data averaged across the remaining 20 participants, the included trials involved RTs that were comparable across contexts; this was true both for $RT_{Short}$ (1,414 ± 91 and 1,422 ± 72 ms in hasty and cautious contexts, respectively) and for $RT_{Long}$ trials (2,030 ± 34 and 2,034 ± 29 ms in hasty and cautious contexts, respectively). The figure also indicates the strong similarity of EMG peak amplitude values for $RT_{Short}$ and $RT_{Long}$ trials and for the hasty and cautious contexts. All individual and group-averaged numerical data exploited for S8 Fig are freely available at this link: https://osf.io/tbw7h. EMG, electromyography; MEP, motor-evoked potential; RT, reaction time; TMS, transcranial magnetic stimulation.
(DOCX)

**S1 Table. (related to Fig 2). The context-dependent shift in SAT did not depend on the session.** As mentioned in the Results section, the session order was not completely counterbalanced among the 50 participants included in the behavioral analysis: 24 participants started the experiment with the hasty session while 26 started with the cautious one. To ensure that the effects of CONTEXT observed on DT, accuracy, urgency intercept reported in Fig 2 did not depend on the lack of counterbalancing, we performed Bayesian rmANOVAs, testing whether any of these effects interacted with the factor SESSION ORDER. We did not find any significant CONTEXT*SESSION ORDER interaction, whether looking at DTs ($F_{1, 48} = 0.00043$, $p = 0.984$, partial $\eta^2 = 8.81 \times 10^{-5}$), at accuracy ($F_{1, 48} = 1.30$, $p = 0.259$, partial $\eta^2 = 0.026$), or at urgency intercepts ($F_{1, 48} = 1.30$, $p = 0.259$, partial $\eta^2 = 0.055$). BFs for these 3 variables were 3.48, 4.45 and 3.22, providing strong evidence for a lack of CONTEXT*SESSION ORDER interaction on these variables. BF, Bayes factor; DT, decision time; rmANOVA, repeated measures analyses of variance; SAT, speed–accuracy trade-off.
(DOCX)

**S2 Table. (related to Fig 4). Decomposition of the CONTEXT*TIMING*REPRESENTATION interaction using separate rmANOVAs for each representation with CONTEXT and TIMING as within-participant factors.** For the sake of homogeneity, the same procedure was applied throughout the manuscript to decompose significant effects following interactions. That is, Tukey HSD post hoc tests were applied for all pairs of conditions comprised in the interaction, and the correction was thus proportional to the number of pairs tested. Here, we exploited an alternative decomposition of the CONTEXT*TIMING*REPRESENTATION interaction found on the chosen side of $TMS_{Finger}$ participants, using 3 separate rmANOVAs for each representation with CONTEXT and TIMING as within-participant factors. First, this analysis showed that the global effect of TIMING reported in the manuscript (Fig 3) could be replicated for the 3 representations using this approach. Further, a CONTEXT*TIMING interaction was present in the index representation ($F_{2,36} = 3.94$, $p = 0.028$, partial $\eta^2 = 0.181$), consistent with the effects reported in the manuscript (Fig 4A, left panel). However, this interaction was not present for the surrounding finger representations, neither for the thumb ($F_{2,36} = 0.44$, $p = 0.646$, partial $\eta^2 = 0.024$) nor for the pinky one ($F_{2,36} = 0.624$, $p = 0.543$, partial $\eta^2 = 0.033$). Interestingly, the thumb representation presented a main effect of CONTEXT (marginally significant: $F_{1,18} = 4.34$, $p = 0.051$, partial $\eta^2 = 0.194$), while this effect was not significant for the pinky representation ($F_{1,18} = 1.82$, $p = 0.194$, partial $\eta^2 = 0.089$). Altogether, this analysis may suggest that the surround suppression effect reported in the manuscript was the strongest in the thumb representation, putatively due to its stronger functional link with the index representation. The analysis also hints that, in the thumb representation, the suppression of excitability did not necessarily depend on the timing at which it was probed and

was potentially already present early on during the decision process (i.e., see Jump$_1$ in Fig 4A). HSD, honestly significant difference; rmANOVA, repeated measures analyses of variance; TMS, transcranial magnetic stimulation.
(DOCX)

**S3 Table. (related to Fig 4). In TMS$_{Finger}$ participants, the effect of context on motor excitability observed on the chosen side did not depend on the session order.** As mentioned in the Results section, the session order was not completely counterbalanced among the 19 TMS$_{Finger}$ participants included in the MEP analysis: 8 participants started the experiment with the hasty session, while 11 started with the cautious one. To ensure that the effects of CONTEXT observed on motor excitability in these participants did not depend on the lack of counterbalancing, we performed Bayesian rmANOVAs, testing whether the factor CONTEXT interacted with SESSION ORDER. We did not find any significant CONTEXT*SESSION ORDER (F$_{1, 17}$ = 2.16, $p$ = 0.159, partial η$^2$ = 0.113), CONTEXT*REPRESENTATION *SESSION ORDER (F$_{2, 34}$ = 1.48, $p$ = 0.240, partial η$^2$ = 0.081), CONTEXT*TIMING *SESSION ORDER (F$_{2, 34}$ = 2.49, $p$ = 0.097, partial η$^2$ = 0.128), or CONTEXT* REPRESENTATION*TIMING*SESSION ORDER interaction (F$_{4, 68}$ = 0.32, $p$ = 0.863, partial η$^2$ = 0.018). BFs for these interactions ranged between 3.08 and 30.09, providing strong to decisive evidence for a lack of effect of the session order on the effect of context on motor excitability. BF, Bayes factor; MEP, motor-evoked potential; rmANOVA, repeated measures analyses of variance; TMS, transcranial magnetic stimulation.
(DOCX)

## Acknowledgments

We thank Sara Lo Presti, Roxanne Weverbergh, and Caroline Hermand for their help in the acquisition of the data.

## Author Contributions

**Conceptualization:** Gerard Derosiere, David Thura, Paul Cisek, Julie Duque.

**Formal analysis:** Gerard Derosiere.

**Funding acquisition:** Gerard Derosiere.

**Investigation:** Gerard Derosiere.

**Methodology:** Gerard Derosiere, David Thura, Paul Cisek, Julie Duque.

**Project administration:** Gerard Derosiere.

**Software:** Gerard Derosiere, David Thura, Paul Cisek.

**Supervision:** Julie Duque.

**Visualization:** Gerard Derosiere.

**Writing – original draft:** Gerard Derosiere.

**Writing – review & editing:** Gerard Derosiere, David Thura, Paul Cisek, Julie Duque.

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
