## [Editor Report · Decision Letter 0]

28 Sep 2021

Dear Dr Derosiere, 

Thank you for submitting your manuscript entitled "Hasty sensorimotor decisions rely on an overlap of broad and selective changes in motor activity" for consideration as a Research Article by PLOS Biology.

Your manuscript has now been evaluated by the PLOS Biology editorial staff, as well as by an academic editor with relevant expertise, and I am writing to let you know that we would like to send your submission out for external peer review.

Please re-submit your manuscript within two working days, i.e. by Sep 30 2021 11:59PM.

Kind regards,

Gabriel Gasque

Senior Editor

PLOS Biology

ggasque@plos.org

---

## [Decision Letter · Decision Letter 1]

16 Nov 2021

Dear Dr Derosiere,

Thank you for submitting your manuscript "Hasty sensorimotor decisions rely on an overlap of broad and selective changes in motor activity" for consideration as a Research Article at PLOS Biology. Your manuscript has been evaluated by the PLOS Biology editors, by an Academic Editor with relevant expertise, and by three independent reviewers. Please accept my apologies for the delay in sending this decision to you.

In light of the reviews (below), we are pleased to offer you the opportunity to address the comments from the reviewers in a revised version that we anticipate should not take you very long. We will then assess your revised manuscript and your response to the reviewers' comments and may consult the reviewers again.

We expect to receive your revised manuscript within 1 month.

Please email us (plosbiology@plos.org) if you have any questions or concerns or would like to request an extension. At this stage, your manuscript remains formally under active consideration at our journal; please notify us by email if you do not intend to submit a revision so that we may end consideration of the manuscript at PLOS Biology.

**IMPORTANT - SUBMITTING YOUR REVISION**

A) Your revisions should address the specific points made by each reviewer. 

B) Please address our editorial requests also listed below my signature. 

C) Please submit the following files along with your revised manuscript:

*Resubmission Checklist*

*Published Peer Review*

*PLOS Data Policy*

*Blot and Gel Data Policy*

Sincerely,

Gabriel Gasque

Senior Editor

PLOS Biology

ggasque@plos.org

EDITORIAL REQUESTS:

1) Please indicate in your manuscript if your protocol approved by the Ethics Committee of the Université Catholique adhered to the principles expressed in the Declaration of Helsinki or any other specific national or international ethical guidelines.

Note that we do not require all raw data. Rather, we ask for all individual quantitative observations that underlie the data summarized in the figures and results of your paper. For an example see here: http://www.plosbiology.org/article/info%3Adoi%2F10.1371%2Fjournal.pbio.1001908#s5

These data can be made available in one of the following forms:

2.1) Supplementary files (e.g., excel). Please ensure that all data files are uploaded as 'Supporting Information' and are invariably referred to (in the manuscript, figure legends, and the Description field when uploading your files) using the following format verbatim: S1 Data, S2 Data, etc. Multiple panels of a single or even several figures can be included as multiple sheets in one excel file that is saved using exactly the following convention: S1_Data.xlsx (using an underscore).

2.2) Deposition in a publicly available repository. Please also provide the accession code or a reviewer link so that we may view your data before publication.

Regardless of the method selected, please ensure that you provide the individual numerical values that underlie the summary data displayed in the following figure panels: Figures 2A-D, 3BC, 4A-C, 5, 6A-C, 7AB, S1CD, S2ABC, S3, S4AB, S5, and S6B.

2.a) Please also ensure that each figure legend in your manuscript includes information on where the underlying data can be found and that your supplemental data file/s has/have a legend.

2.b) Please ensure that your Data Statement in the submission system accurately describes where your data can be found.

REVIEWS:

Reviewer #1: This is an impressive paper. It rarely happens that I have no 'major comments' on a manuscript, but this paper is meticulous in all regards, and the results are very interesting and convincing. It is very well written and suited for this journal as well. My only minor comments are:

Line 206: is the word 'in' missing here?

A reference to Figure 6.C seems to be missing

Line 822: should 30% be 40%?

Line 896: change "between each block" to "between blocks"?

Line 1069-1070: "…, we determined the level of sensory evidence at the time of commitment (i.e., at DT)." This did not make sense to me, as DT is a duration and not a time point. If I understand things correctly, then the time point that the authors refer to here is [time of jump1] + [time of sensory processing] + DT

Reviewer #2: Derosiere et al. present data from three samples of participants who made left or right index finger responses based on the slow accumulation of tokens to the left or right of a computer monitor. All participants completed two types of block, varying the punishment for incorrect responses in order to manipulate their internal speed-accuracy tradeoff settings. Two of three groups received concomitant TMS, targeting either the hand or leg motor representation, with MEPs recorded in each case from 3 muscles at four possible times. This provided a somatotopically broad assay of motor activity as pressure (and to a lesser extent evidence) built towards a response. Key findings were a general increase in corticospinal excitability through the trial, with an increased focus on speed being associated with further enhanced excitability of the responding finger and ipsilateral leg, but reduced excitability of ipsilateral task-irrelevant digits. Increased focus on speed was also associated with a reduction in the correlation between MEPs of the responding finger and other digits, although the statistical inference for the latter result is unconventional (and less demanding than is traditional).

This is a well-written paper. In terms of the novelty of the result - I cannot claim anything like exhaustive knowledge of this field, but I know that at least one previous publication (which the authors cite) has investigated how corticospinal excitability (measured via TMS/MEPs) varies across the period of motor preparation alongside a manipulation of the speed-accuracy tradeoff. However, the current work differs from this (and many other) TMS/MEP papers by measuring a large number of task-irrelevant muscles (both close to and far from the task relevant ones) to probe the motor cortex more widely. This is an appealing feature. I personally find the result interesting, and there are implications for neurocomputational theories of action preparation. I imagine there would be some interest both within and outside the immediate field. From a methodological perspective, I have a few queries, but the work seems broadly solid and the conclusions are sensible.

Essential:

1) The authors' final analysis investigates correlations between MEPs from the responding index finger and other muscles of the body. They then compare these correlations between SAT contexts, finding some significant differences. The problem is that the statistical inference for these results is unconventional, and readers are likely to take them to mean something that they do not. To generate their results, the authors pool MEPs across multiple participants, which violates the assumption of independent data points. This is unlikely to generate spurious correlations, as the MEPs are normalised for each individual before being combined. However, the associated p values (for both the correlations, and the subsequent contrast across contexts) will not mean what most readers will take them to mean. When confronted with a p value from a group analysis, we expect that the p value permits a generalisation from the study sample to the population of all people (or at least all people with similar characteristics to the sample). This is not currently the case for this analysis, and p values will be lower than they would be for the standard generalisation. Instead, the p values reported here reflect a generalisation to all MEPs coming specifically from this particular set of people. In approximate order of satisfactoriness, possible solutions would be: a) running this analysis using the correct multilevel model; b) calculating the correlations for each individual and then comparing the group-average correlations across contexts in a second step; or c) explaining to the reader in the results and/or discussion that p values (and associated significance) are exaggerated relative to a standard group-level statistical analysis.

2) The authors perform a fairly elaborate data-matching procedure in order to ensure that RTs were similar for trials included from the two SAT contexts. I understand the concern over matching nearness to the ultimate motor act, but, unless I have misunderstood, I think the procedure could replace one concern with another. Specifically, the adjusted data essentially compares slow trials from the high-pressure condition with fast trials from the low-pressure condition. Despite the fact that evidence is maintained within a fairly ambiguous range across the TMS period (which, by the way, should be emphasised in the results, and not just in the detailed methods) it seems to me that a relative difference in speed of response might reflect systematic differences in the evidence (i.e. number of tokens) accumulated earlier on. Evidence might therefore not be well matched (through randomisation of trials) as it would be in the unselected data. This concern would apply particularly to later TMS times. It matters, because the authors wish to tie their results as far as possible to changes in SAT context (and therefore urgency). Ideally, I would like to see the authors report on the average evidence in favour of the responding digit for different times/contexts based specifically on the selected subsets of trials. Furthermore, if there is any evidence that it differs, I think it would be beneficial to report whether the same results are obtained based on the full (unmatched) data, even if only briefly, e.g. a few sentences linked to a supplementary figure.

3) For the most important experimental manipulation (SAT context) we are told that the order of the sessions was counterbalanced across participants (line 868). However, this cannot be completely true for two out of three groups (which have odd numbers of participants), and, in particular, this may be more substantially incorrect for the MEP analyses, from which a number of participants have been excluded. Failure to fully counterbalance introduces a systematic confound across the main experimental manipulation. Solutions, in approximate order of satisfactoriness, would include: a) increasing the sample to make sure that counterbalancing is complete for all samples from which key results are derived; b) reporting the actual degree of counterbalancing and including block order as a factor in Bayesian ANOVAs to provide evidence favouring the null for associated main effects and interactions; c) reporting the actual degree of counterbalancing and discussing this weakness.

4) It may be an oversight on my part, but I saw no explicit mention of the randomisation of the order of trials for non-blocked factors such as TMS timing. This was almost certainly applied, but should nonetheless be clearly stated (as if this was not applied it would be a problematic systematic confound).

5) Key results are analysed using three-factor ANOVAs. I don't have any serious doubts about these results (although I personally would have started from a four-factor ANOVA including response side, in order to make a clearer statistical inference regarding the different patterns obtained for responding and non-responding sides). However, the pipeline for following up interactions from a high-dimensional ANOVA seems non-standard (pg. 13). The authors go straight from a three-way interaction to pairwise t-tests (with Tukey correction, but no indication of how many contrasts are being considered in this correction procedure). A more standard way to follow up the context*timing*representation interaction, based on the way the authors choose to organise the data in Figure 4, would be to next perform three separate two-way (context*timing) ANOVAs, one per digit. It looks likely that a two-way interaction would then be found for the responding index digit, to be followed up by three pairwise tests to reveal the important difference between contexts at token 7 (so no real change there). However, it's less clear that such a 2x3 interaction would emerge for the pinky or thumb. Without it, the pairwise follow ups would be unnecessary and the final result would (most likely) be a main effect of context, i.e. an effect on average across all three times, for these digits. The authors may not like my suggestion to follow a more standard follow-up pipeline, but they should at least state exactly how many (and which) pairwise tests were run and corrected for. They should also make a clear statement that all other main effects and interactions from their three-way ANOVAs were non-significant (or provide a bit more information about this if that is not the case, e.g. the way in which the presentation has been made selective for the purpose of narrative clarity). 

Additional:

Page 9 - the authors assess the behavioural SAT in the entire sample, which is reasonable. However, they go on to view effects on corticospinal excitability in a selected sample that is not likely to be random (relative to the overall sample) in terms of its experience of a SAT. Specifically, it seems likely that the participants who get excluded (as a result of RT matching) would mainly be those who have the largest behavioural SAT. I would therefore suggest that the behavioural results demonstrating a SAT be run in just the included samples from later analyses, as a supplementary figure or similar.

"251 Further, a Bayes Factor analysis provided evidence for a lack of effect of the subgroup on all of these behavioral data."

I appreciate that full stats are provided in the supplementary materials, but it feels odd not to have anything at all here (e.g. the range of Bayes factors observed). More generally, for the Bayesian statistics, the authors should say something about the assumptions made regarding the alternative hypothesis (e.g. what was the prior on effect size) or at least identify the software/routine they used, stating that default settings were used if that is the case.

"288 Normalized MEP amplitudes displayed a main effect of TIMING in TMSFinger subjects (i.e., Jump1 vs Jump4 vs Jump7 in a repeated-measures [rm]ANOVA)."

It would be helpful to the reader (mainly for anticipating what is to come) to start out by stating that this result comes from a three-way ANOVA with factors timing, context and representation (although I appreciate that this info appears in the methods). 

Discussion section. The authors don't seem to have pointed out any weaknesses or limitations to their work. While understandable (if not condonable) from a strategic perspective, this is poor scientific practice.

Lines 545-556. Is it also worth considering functional/anatomical coupling between muscles? This would seem a particular feature of the motor system compared to sensory systems (i.e. to generate one action we almost never utilise a single muscle in isolation, although that would be more plausible here than in more natural contexts requiring stabilising postural responses and so forth).

Methods section - it would be helpful to say something about the basis for selecting the size of the three samples. It would also be helpful to say something about the degree to which analyses were developed a priori vs. post hoc. For example, in my experience, in the absence of pre-registration it is typical to have a broad sense of the intended analyses (e.g. mean differences in MEPs will be assessed via ANOVA) without precisely mapping out all the steps (e.g. the thresholds for exclusion of trials/subjects; the factors included in the ANOVAs vs. treated in separate ANOVAs).

The information about the number of trials of each kind (e.g. pg. 36) is rather difficult to digest. Could the authors present a table to help with this, e.g. showing the number of trials per block for each intersection of design-level factors? 

1157 "In order to prevent contamination of the measurements from background muscular activity, trials in which the root mean square of the EMG signal exceeded 3 SD above the mean before stimulation (i.e., -250 to -50 ms from the pulse) were discarded from the analyses (rejection rate: 8.48 ± 0.43 % in TMSFinger subjects and 0.99 ± 0.05 % in TMSLeg subjects)."

Would this procedure not leave participants with preactivation on the majority of trials relatively untouched? Were there any such participants?

1255 "To this aim, we exploited the single-trial MEPs obtained at Jump7 following the 100 iterations of the RT-matching procedure described above. We first normalized single trial MEPs as a percentage of the average baseline amplitude for each finger representation, each timing and each context and then computed Pearson's correlations between the five pairs of muscles for each timing and each context."

Is there a typo here? If not, how can this have been done for each timing when only jump7 trials are included?

Reviewer #3: This study tests the hypothesis that speed-accuracy tradeoff (SAT) relies on influences exerting broad changes on the motor system - i.e., increasing the activity of the motor system globally in hasty actions.

Fifty subjects were trained to perform a perceptual decision task, in which 15 tokens jump one-by-one every 200 ms from a central circle to one of two lateral target circles. The subjects had to choose which

of the targets will receive the majority of tokes by pressing a keyboard key using left (for the left target) or right (for the right target) index finger. Subjects were free to respond at any time from Jump-1 to Jump-15.

Correct trials led to reward, whereas incorrect trials to penalties. The reward decreased over the course of the trial, producing an increasing urge to decide. In one block of trials, incorrect decisions were severely penalized

(-14 cents), emphasizing the need for cautiousness. However, on another block of trials, the penalty of incorrect decisions was only -4 cents, encouraging subjects to make hasty decisions. Transcranial Magnetic Stimulation (TMS)

was applied on multiple motor representations of the motor cortex to elicit motor evoked potentials (MEP) in fine finger and leg muscles. The results showed that hasty decisions cause a broad amplification of motor activity but

only to the chosen side. Importantly, the motor activity of the areas surrounding the choses index representation was suppressed in hasty decisions. This is an interesting study that explores a very important topic in motor

control, and fits within the scope of PloS Biology. However, there are some issues that the authors need to address before the manuscript is considered for publication

Major issues 

1. The study uses TMS to elicit motor evoked potential (MEP) in nine finger and leg muscles. There are many studies that report strong variability of the MEP responses induced by TMS [1]. 

Although this variability can be generated by technical factors, such as the location, orientation and stability of the coils, this variability remains even after controlling these factors.

Some studies showed that this inherent variability can result from neurophysiological changes in the CSE pathway. Some studies remove the first few MEPs, since most of the variability is 

expected in the first few MEPs due the regional changes of the blood flow [3], although current study argues that even removing the first few MEPs does not further increase the reliability of MEP responses.

So, how did the author deal with this inherent pitfall of the TMS to induce MEP responses? 

2. One of the key findings of this study is that the hasty context was associated with particularly large MEPs, including in the leg muscles, though this effect was limited to the chosen side.

So, if hasty actions affect both fingers and leg muscles, it could be argued that the effects are related to the attentional mechanisms - i.e., hasty actions require increase of attention for selecting the correct options.

In other words, since the effects are non-effector specific, then we cannot exclude the hypothesis that large MEPs are associated with increase on the participants' attention. The authors need to discuss this 

alternative hypothesis.

3. How can the affordance competition hypothesis, which was proposed by Paul Cisek (one of the authors of the manuscript) explain the effects of the context decision (hasty vs. caution) to the MEPs?

4. Is it possible the TMS to cause any changes on the decision-making process? Previous studies have showed that "temporal lesions" from TMS can bias the decision or change reaction time and other characteristics

of the decision-making process. How does the author ensure that TMS did affect the decision process?

5. How did they ensure that they targeted the same motor areas with the same direction across all subjects? 

Minor issue

1. How did they select the four different time instance (baseline, 1, 4, 7) to perform the TMS? What was the criterion?

2. Was it any difference on the MEP responses between correct and incorrect trials?

[1] Current orientation induced by magnetic stimulation influences a cognitive task. Hill AC, Davey NJ, Kennard C Neuroreport. 2000 Sep 28; 11(14):3257-9.

[2] A checklist for assessing the methodological quality of studies using transcranial magnetic stimulation to study the motor system: an international consensus study.

Chipchase L, Schabrun S, Cohen L, Hodges P, Ridding M, Rothwell J, Taylor J, Ziemann U Clin Neurophysiol. 2012 Sep; 123(9):1698-704.

[3] Cortical hemoglobin-concentration changes under the coil induced by single-pulse TMS in humans: a simultaneous recording with near-infrared spectroscopy.

Mochizuki H, Ugawa Y, Terao Y, Sakai KL Exp Brain Res. 2006 Mar; 169(3):302-10.

---

## [Decision Letter · Decision Letter 2]

3 Feb 2022

Dear Dr Derosiere,

Thank you for submitting your revised Research Article entitled "Hasty sensorimotor decisions rely on an overlap of broad and selective changes in motor activity" for publication in PLOS Biology. I have now obtained advice from original reviewers 2 and 3 and have discussed their comments with the Academic Editor. 

Based on the reviews, we will probably accept this manuscript for publication, provided you satisfactorily address the remaining points raised by reviewer 2. The Academic Editor thinks that the statistical concern is justified and that it would be the preferable solution if you could fix it with a multilevel statistical model to distinguish within from between subject variance, or else work around it with one of the strategies suggested by the reviewer. 

Please also make sure to address the following data and other policy-related requests:

1) Please indicate in your manuscript if your protocol approved by the Ethics Committee of the Université Catholique adhered to the principles expressed in the Declaration of Helsinki or any other specific national or international ethical guidelines.

Note that we do not require all raw data. Rather, we ask for all individual quantitative observations that underlie the data summarized in the figures and results of your paper. For an example see here: http://www.plosbiology.org/article/info%3Adoi%2F10.1371%2Fjournal.pbio.1001908#s5

These data can be made available in one of the following forms:

2.1) Supplementary files (e.g., excel). Please ensure that all data files are uploaded as 'Supporting Information' and are invariably referred to (in the manuscript, figure legends, and the Description field when uploading your files) using the following format verbatim: S1 Data, S2 Data, etc. Multiple panels of a single or even several figures can be included as multiple sheets in one excel file that is saved using exactly the following convention: S1_Data.xlsx (using an underscore).

2.2) Deposition in a publicly available repository. Please also provide the accession code or a reviewer link so that we may view your data before publication.

Regardless of the method selected, please ensure that you provide the individual numerical values that underlie the summary data displayed in the following figure panels: Figures 2A-D, 3BC, 4A-C, 5, 6A-C, 7AB, S1CD, S2ABC, S3, S4AB, S5, and S6B.

2.a) Please also ensure that each figure legend in your manuscript includes information on where the underlying data can be found and that your supplemental data file/s has/have a legend.

2.b) Please ensure that your Data Statement in the submission system accurately describes where your data can be found.

We expect to receive your revised manuscript within two weeks. 

*Published Peer Review History*

*Early Version*

Sincerely,

Gabriel

Gabriel Gasque, Ph.D.,

Senior Editor,

ggasque@plos.org,

PLOS Biology

Reviewer remarks:

Reviewer #2: The authors have responded appropriately to the majority of my concerns and the paper is nearly ready for publication. However, I need them to make a further adjustment in relation to essential point #1 from my original review.

The authors have opted to address this point by adding a caveat paragraph in the discussion. Most of this is fine, but they conclude it with (line 680):

"However, this procedure was applied for both contexts and the putative exaggeration of statistical values, if any, should have thus concerned both contexts. Despite this, we observed differences in R-values between contexts, which represent the most important findings of this whole correlation analysis."

Unfortunately, the issue I described relates to the p values for both the correlations themselves, and the test of differences between correlations (i.e. Fisher's z procedure). Fisher's z procedure produces a statistic based on r that is normally distributed (and thus suitable for a subsequent t-test) but the standard error of this statistic (i.e. the essential component that normalises group differences to determine p values) is determined using the sample size (N) under the assumption of independent data points. The authors are, I assume, providing an N of 367 for this calculation (or, more probably, giving their software 367 rows of data so that it will assume a sample size of 367). But of course, they only really have 19 participants. A standard error based on a sample size of 367 will be about four times smaller than one based on a sample size of 19, yielding, for example, a t statistic that is about four times larger. This is what I mean when I say they underestimate their p values (in relation to a typical generalisation to a population of all people). As I mentioned in my original review, the best solution would be to use a multilevel model or two-step procedure that will yield p values with the standard meaning. If this is not done, the authors need to be unequivocal in describing this limitation (or else explain how/why I have misunderstood their Fisher's z procedure in their reply).

Reviewer #3: The authors have successfully addressed all my concerns/comments. I recommend this study for publication to the PLOS journal

---

## [Editor Report · Decision Letter 3]

10 Mar 2022

Dear Dr Derosiere,

On behalf of my colleagues and the Academic Editor, Alexander Gail, I am pleased to say that we can in principle accept your Research Article "Hasty sensorimotor decisions rely on an overlap of broad and selective changes in motor activity" for publication in PLOS Biology, provided you address any remaining formatting and reporting issues. These will be detailed in an email that will follow this letter and that you will usually receive within 2-3 business days, during which time no action is required from you. Please note that we will not be able to formally accept your manuscript and schedule it for publication until you have any requested changes.

PRESS

Sincerely, 

Richard

Richard Hodge, PhD

Associate Editor, PLOS Biology

rhodge@plos.org

On behalf of:

Gabriel Gasque, PhD 

ggasque@plos.org